# Study on landslide susceptibility mapping with different factor screening methods and random forest models

**Tengfei Gu**[1,2], **Jia Li**[1]*, **Mingguo Wang**[3], **Ping Duan**[1], **Yanke Zhang**[4], **Libo Cheng**[1]

**1** Faculty of Geography, Yunnan Normal University, Kunming, Yunnan Province, China, **2** Badong National Observation and Research Station of Geohazards, China University of Geosciences (Wuhan), Wuhan, Hubei Province, China, **3** Yunnan Institute of Geological Surveying and Mapping Company Limited, Kunming, Yunnan Province, China, **4** Wuhan Tianjihang Information Technology Company Limited, Wuhan, Hubei Province, China

* keguigiser@163.com

**Data Availability Statement:** Data Availability Statement: Some of the original data used in this paper are confidential, including the landslide map and 1: 200,000-scale geological map, we can't

## Abstract

The number of input factors affects the prediction accuracy of a model. Factor screening plays an important role as the starting point for data input. The aim of this study is to explore the influence of different factor screening methods on the prediction results. Taking the 2014 landslide inventory of Jingdong County as an example, a landslide database was constructed based on 136 landslide events and 11 selected factors, which were randomly divided into a training dataset and a test dataset according to a ratio of 7:3. Four factor screening methods, namely, the information gain ratio (IGR), GeoDetector, Pearson correlation coefficient and multicollinearity test (MT), were selected to screen the factors. A random forest (RF) model was then used in combination with each factor set for landslide susceptibility mapping (LSM). Finally, accuracy validation was performed using confusion matrices and ROC curves. The results show that factor screening is beneficial in improving the accuracy of the resulting model compared to the original model. Second, the IGR_RF model had the highest AUC value (0.9334), which was higher than that of the MT_RF model without factor screening (0.9194), and the IGR_RF model predicted the most landslides in the very high susceptibility zone (51.22%), indicating the good prediction performance of the IGR_RF model. Finally, the factor weighting analysis revealed that NDVI, elevation and aspect had the greatest influence on landslides in Jingdong County and that curvature had the least influence on landslides. This study can provide a reference for factor screening in LSM.

## 1. Introduction

Landslides are a common geological hazard in mountainous areas and are characterized by wide distributions, high frequencies of occurrence and rapid hazard formation [1]. Jingdong County is near the collision zone between the Indian Ocean plate and the Eurasian plate, exhibits active crustal movement, and is one of the key landslide prevention areas in Yunnan Province [2]. In 2008, a landslide and other natural hazards occurred in Taizhong township,

share them publicly online. Landslide data from the Yunnan Bureau of Geology and Mineral Resources (http://www.yndkch.com), 1: 200,000-scale geological map from the National Geological Archives of China (http://www.ngac.org.cn/DataSpecial/geomap.html). Researchers who meet the criteria for access to confidential data may apply from the above institutions. All other data can be downloaded through the download link in the manuscript.

**Funding:** J.L., Grant No. 41961061, National Natural Science Foundation of China, http://www.nsfc.gov.cn/. J.L., Grant No. 202001AT070057, 202301AT070061, Yunnan Fundamental Research Projects, http://kjt.yn.gov.cn/. J.L., P.D., Grant No. YNWR-QNBJ-2020-103, YNWR-QNBJ-2020-048, 'Revitalizing Yunnan Talents Support Program' project funding support, https://www.yn.gov.cn/. J. L. Grant No. 202105AC160059, Reserve Talent Program for Young and Middle-aged Academic and Technical Leaders in Yunnan Province, http://kjt.yn.gov.cn/. The funders had no role in study design, data collection and analysis, decision to publish, or preparation of the manuscript.

**Competing interests:** The authors have declared that no competing interests exist.

Jingdong County, affecting 7,604 people in the township [3]. In 2016, 1 person died and more than 3,600 people were affected by landslides and other geological hazards in Jingdong County, with direct economic losses of approximately RMB 8.115 million yuan [4]. How to effectively prevent and reduce the occurrence of new landslide hazards is one of the main points of work for government departments today. Landslide susceptibility mapping (LSM) is a common method for landslide risk assessment [5]. LSM is the use of mathematical language to describe the probability of future landslides in an area under the influence of topography and geomorphology, geological structure, hydrology and meteorology, human activities and other predisposing factors, and it is used to map the distribution of landslide susceptibility [6, 7]. LSM can provide a scientific basis for land planning, support major construction site selection and monitoring of landslide hazards, protect people's lives and property effectively, and reduce unnecessary losses [8].

With the rapid development of data mining technology, machine learning models have become a popular method in LSM due to their good predictive performance [9, 10]. Machine learning models generate classifiers by learning attribute features of known landslides and then use the classifiers to make predictions for the entire study area. Such models include decision trees [11, 12], random forests [13–15], support vector machines [16, 17], naive bayes [18, 19], artificial neural networks [20, 21], etc. In some machine learning model comparisons, random forest (RF) models were demonstrated to have better predictive power than other models in several study areas [22–25]. The RF model is a type of bagging ensemble algorithm that uses multiple decision trees to construct a model and reduces the risk of overfitting through the principle of averaging, yielding a high prediction accuracy and strong generalization ability [26].

However, for machine learning models, factor screening [27, 28], sample selection [29], mapping units [30, 31], evaluation models [32], and optimization of model parameters [33] can affect the final prediction accuracy. Factor screening plays an important role as the starting point for data input. Through factor screening, selecting factors with a high correlation to landslides can improve the performance of the model and reduce redundant information and noise. Additionally, factor screening can reduce the dimensionality of the data and reduce the complexity of the model, thus reducing the risk of overfitting and improving the robustness of the model. Finally, factor screening can speed up model training time and improve efficiency. In contrast to the method of qualitative factor screening based on expert experience, researchers tend to use a quantitative approach to select factors. This is because the main factors for landslide generation are not exactly the same in different areas [34].

Currently, factor screening methods include the multicollinearity test, Pearson correlation coefficient, GeoDetector, information gain ratio, recursive feature elimination, rough set, frequency ratio, deterministic coefficient, etc [35–40]. Luo [41] et al. used a multicollinearity test and information gain ratio to evaluate the predictive power of factors and eliminated three factors. Wang [42] et al. eliminated 8 factors based on the correlation coefficient analysis and neighborhood rough set methods. Sun [43] et al. identified and eliminated 7 less important factors using GeoDetector. Wen et al. [44] used recursive feature elimination to eliminate 14 factors, achieving better prediction accuracy with fewer factors. The factors selected by different factor screening methods differ. Therefore, it is worth exploring which factor screening method is beneficial to improving the predictive performance of a model before proceeding with LSM. Based on the existing factor screening methods, four methods were selected for comparative analysis: information gain ratio (IGR), GeoDetector (GD), Pearson correlation coefficient (PCC) and multicollinearity test (MT). These screening methods can be divided into two categories according to the types of factors: reclassified factor data and raw (normalized) factor data. Methods for using reclassified factor data include IGR and GD. The

relationship between the factors and landslides is determined based on the frequency or variance in landslide occurrence in each factor subclass classification, and then factor screening is performed using the factor weights. This type of method focuses on describing the quantitative relationship between factors and landslides and can give the weight of each factor. Methods using raw (normalized) factor data include PCC and MT. The mathematical relationships between the calculated data are determined based on one-to-one or one-to-many factors and then screened according to the magnitude of the correlation between the factors. This type of method focuses on describing the correlation among the different factor data and does not assign weights to the factors.

Jingdong County is located in the southwestern part of Yunnan Province, China, and features complex terrain and frequent geological hazards that threaten the lives and property of the people. Based on the landslide hazard data of Jingdong County, four methods, namely, IGR, GD, PCC and MT, were used for factor screening. Then, they were combined with RF models for LSM, and accuracy validation and analysis were performed. The objectives of this study are (1) to explore the effects of factor screening on LSM through comparative analysis, (2) to find the best factor screening method through comparative analysis, and (3) to provide a reference for other scholars conducting factor screening.

## 2. Study area and data

### 2.1. Study area

Jingdong County is located in southwestern Yunnan Province and the northern end of Puer city. Its geographical location is from longitudes 100°22′E to 101°15′E and latitudes 23°56′N to 24°50′N, covering an area of 4,532 km² (Fig 1). The elevation range is between 778 and 3344 m and the terrain is highly undulating. The county has 10 towns and 3 townships under its

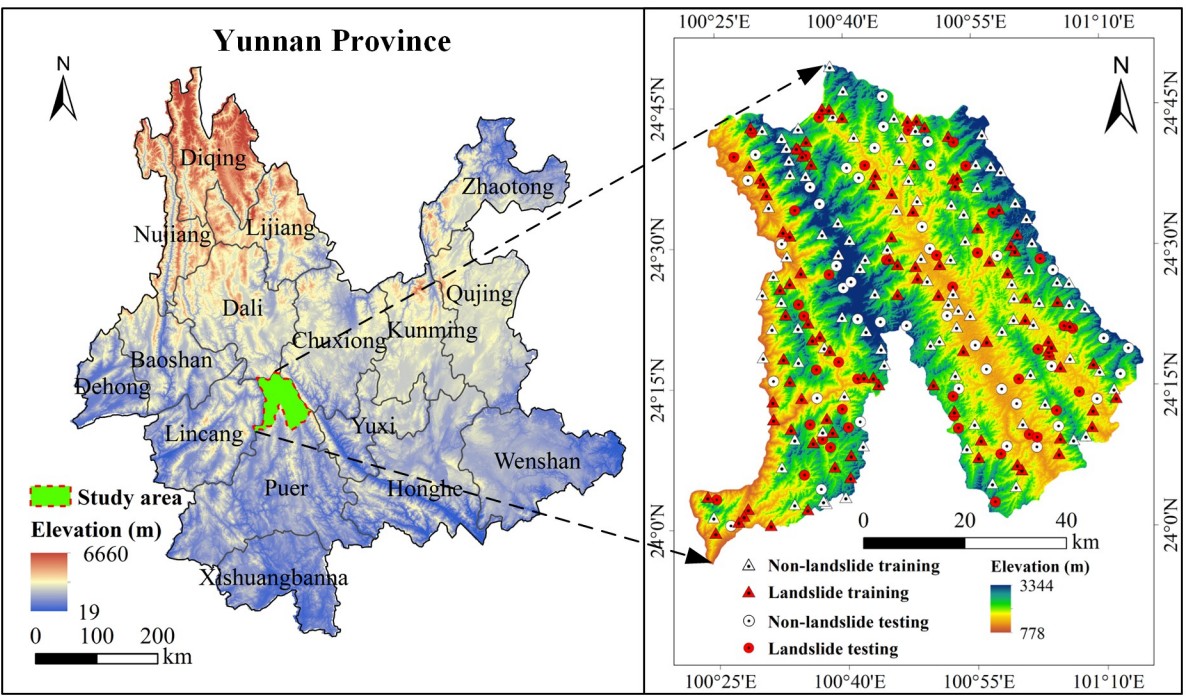

**Fig 1. The study area and landslides.**

jurisdiction. Jingdong County is located at the southern end of the Hengduan Mountains. The main mountains in the study area are the Wuliang Mountain System and the Ailao Mountain System. Due to effects from geological fractures, uplift, erosion, denudation, and sedimentation, a mountainous landform with alternating mountains and valleys has been formed. The terrain is rough in the north and smooth in the south. In addition, influenced by the Indian Ocean plate and the Asian-European plate, the crustal activity is intense, making it one of the key areas for geological hazard prevention in Yunnan Province. The strata exposed in the study area mainly include the lower Cretaceous ($K_1$), the middle Jurassic ($J_2$), the upper Triassic ($T_3$), the upper Jurassic ($J_3$) and the lower Permian ($P_1$). The rivers in the study area belong to the Lancang River system and the Red River system, and the area has a subtropical monsoon climate. The average annual rainfall and annual temperature are 1086.7 mm and 18.3°C respectively. It has distinct dry and wet seasons. Jingdong County is the only county in China to have two national nature reserves—the Wuliang Mountain and Ailao Mountain National Nature Reserves—and is rich in forest resources, with a county-wide forest coverage rate of 77.01%.

## 2.2. Landslide data sources and processing

The data on landslides in Jingdong County are based on interpretations from remote sensing images of geological hazards in Jingdong County in 2014. The landslide data were geolocated, and 136 landslide points were obtained. The slope unit was chosen as the evaluation unit. The slope unit divides the study area into many large and small slopes according to the basic shapes of the mountains and connects hazard formation conditions with natural factors such as topography and geomorphology. The slope unit is more suitable than the grid for LSM. In this paper, we used the curvature watershed method to divide slope units, resulting in 12,785 slope units in total [45]. Based on the slope units where the landslide points were located, 136 landslide units were selected as the modeling data. In addition, 136 non-landslide units were randomly selected outside 500 m buffer zones around the landslide units [46]. A total of 272 sample data were obtained; 70% (190) of the slope units were used as training data, and the remaining 30% (82) were used as test data [47].

## 2.3. Factor data sources

The formation of landslides is restricted by a variety of internal and external forces [48]. The dominant factors affecting landslides vary among study areas. A total of 11 factors were selected according to field investigation and a review of the relevant literature. The selected factors combined characteristics such as data accessibility and quality of acquired data and included elevation, slope, aspect, curvature, average annual precipitation (AAP), normalized difference vegetation index (NDVI), topographic wetness index (TWI), distance to roads (DTRO), lithology, distance to faults (DTF) and distance to rivers (DTRI). The factor data sources and processing steps are shown in Table 1. The processed factors were converted into a raster data format with a resolution of 30 m×30 m and unified under the same projection coordinate system, and the results are shown in Fig 2.

## 2.4. Factor preprocessing

The slope unit was used as the basic evaluation unit. Each slope unit contained many grid cells, but there should be only one attribute value. Therefore, different statistical methods were used to assign values to the slope units for different types of factors. For continuous factors (such as elevation and slope), the mean value of all the grid cells in a slope unit was considered the attribute value; for discrete factors (aspect and lithology), the mode of all the grid cells in a slope

**Table 1. The factor data sources and processing.**

| Factor | Data source and processing | Data type | Data structure | Data accuracy |
|---|---|---|---|---|
| Elevation | Dataset provided by the USGS National Map Viewer (http://viewer.nationalmap.gov/viewer/) (Fig 2a). | Continuous | Grid | 30 m |
| Slope | Calculated from elevation data (Fig 2b). | Continuous | Grid | 30 m |
| Curvature | Calculated from elevation data (Fig 2c). | Continuous | Grid | 30 m |
| AAP | The data were provided by the Resource and Environmental Science Data Center of the Chinese Academy of Sciences (http://www.resdc.cn/) (Fig 2d). | Continuous | Grid | 1,000 m |
| NDVI | NDVI was obtained from Landsat 8 operational land imager (OLI) images [49] (Fig 2e). Landsat 8-OLI data were provided by the USGS National Map Viewer (http://viewer.nationalmap.gov/viewer/). | Continuous | Grid | 30 m |
| TWI | Determined based on water flow direction and aspect data [50] (Fig 2f). | Continuous | Grid | 30 m |
| DTRO | Road data were downloaded from OpenStreetMap (https://www.openstreetmap.org). DTRO was obtained by calculating Euclidean distances to the road data (Fig 2g). | Continuous | Grid | 30 m |
| DTF | Obtained by calculating the Euclidean distance of fault data from a 1:200,000 geological map (Fig 2h). | Continuous | Grid | 30 m |
| DTRI | River data were obtained from the Yunnan Bureau of Geology and Mineral Resources. DTRI was obtained by calculating Euclidean distances to the river data (Fig 2i). | Continuous | Grid | 30 m |
| Aspect | Calculated from elevation data (Fig 2j). | Discrete | Grid | 30 m |
| Lithology | Lithology data were obtained from the 1:200,000 geological map of China (Fig 2k). | Discrete | Vector | 1:200,000 |

unit was considered the attribute value. Different data types were used for different factor screening methods; therefore, different methods were applied to process the factors in the experiment. The IGR and GD methods required reclassified factors, so the factors were reclassified. All the factors except aspect and lithology were classified at equal intervals [51]. The classification criteria are shown in Table 2. The PCC and MT methods required the original factor data to be normalized to eliminate the problem of nonuniform dimensions between factors [52]. The best factors were obtained with the factor screening methods; then, those normalization factors were introduced into the RF models.

## 3. Methods

The slope unit was used as the basic evaluation unit. First, data on the historical landslides and factors in Jingdong County were preprocessed. Non-landslide points were identified based on the landslide point data and were selected at a 1:1 ratio. A landslide database was constructed according to the relationship between the sample points and factors, and the data in the landslide database were divided into training data and test data at a 7:3 ratio. Second, the training data were introduced separately into the four factor screening methods, and the best factors were calculated according to the different methods. Finally, the four groups of factors determined to be the most suitable were introduced (as normalized data) into the RF model for LSM, and then the results were analyzed. A flowchart of the research method is shown in Fig 3.

### 3.1. Factor screening methods

Choosing appropriate factors for LSM is beneficial to optimize the prediction accuracy of the RF model. Among the various factor screening methods, the optimal combination of factors varies with the data and algorithms used. Factor data types include reclassified factors and original (normalized) factors. The methods that require reclassified factors include the IGR and GD; the methods that require original (normalized) factors include PCC and the MT. Therefore, to compare the effects of different factor screening methods on the RF model, in this paper, four factor screening methods were combined with an RF model.

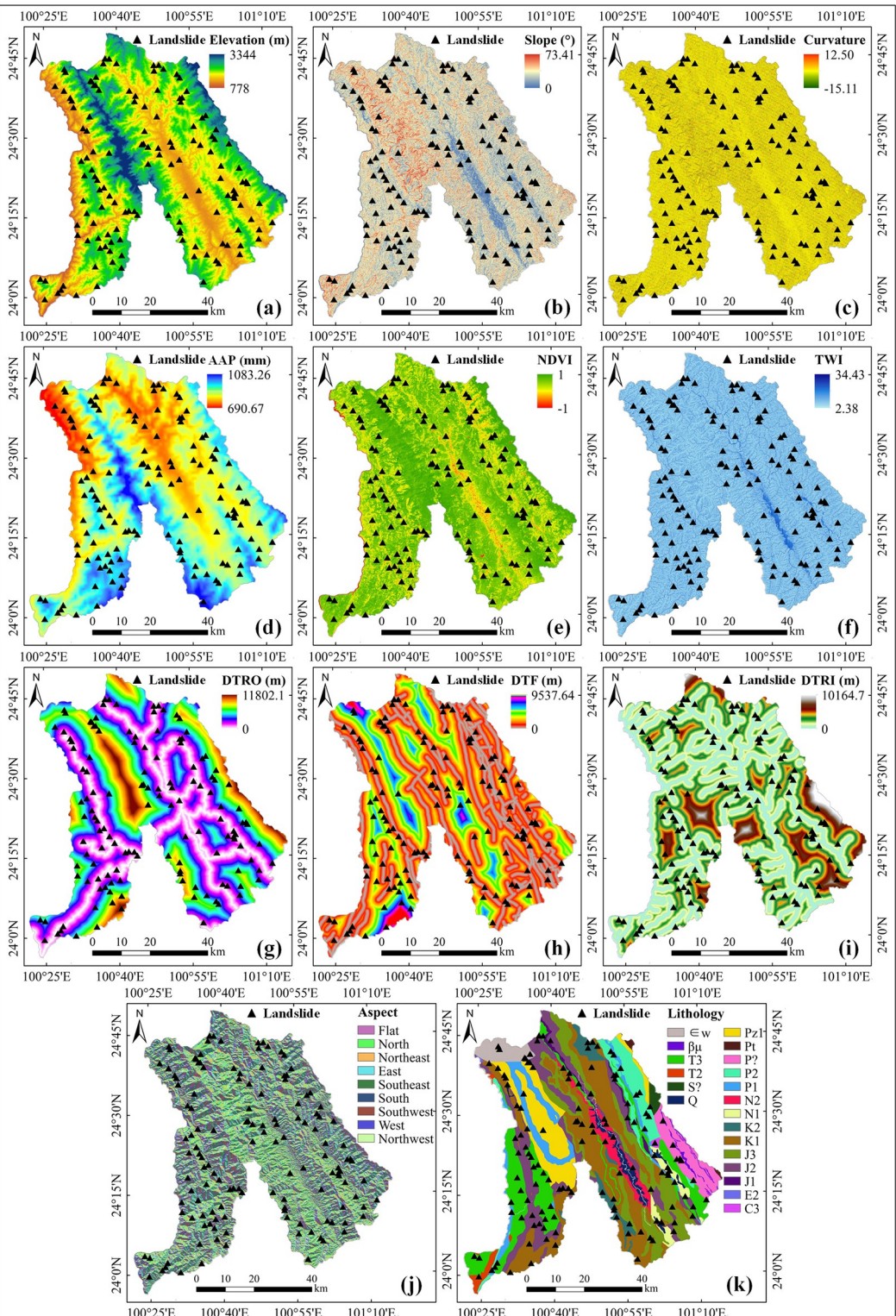

**Fig 2.** Thematic map of landslide factors: (a) elevation, (b) slope, (c) curvature, (d) average annual precipitation (AAP), (e) normalized difference vegetation index (NDVI), (f) topographic wetness index (TWI), (g) distance to roads (DTRO), (h) distance to faults (DTF), (i) distance to rivers (DTRI), (j) aspect, and (k) lithology.

**Table 2. Factor classification descriptions based on slope units.**

| Factor | Classification | Description |
|---|---|---|
| Elevation (m) | 10 | <950, 950–1200, 1200–1450, 1450–1700, 1700–1950, 1950–2200, 2200–2450, 2450–2700, 2700–2950, >2950 |
| Slope (˚) | 10 | <8, 8–12, 12–16, 16–20, 20–24, 24–28, 28–32, 32–36, 36–40, >40 |
| Curvature | 5 | <-0.31, -0.31 to -0.12, -0.12–0.06, 0.06–0.25, >0.25 |
| AAP (mm) | 7 | <750, 750–800, 800–850, 850–900, 900–950, 950–1000, >1000 |
| NDVI | 6 | <0, 0–0.2, 0.2–0.4, 0.4–0.6, 0.6–0.8, >0.8 |
| TWI | 7 | <5.5, 5.5–6.5, 6.5–7.5, 7.5–8.5, 8.5–9.5, 9.5–10.5, >10.5 |
| DTRO (m) | 12 | <1000, 1000–2000, 2000–3000, 3000–4000, 4000–5000, 5000–6000, 6000–7000, 7000–8000, 8000–9000, 9000–10000, 10000–11000, >11000 |
| DTF (m) | 9 | <1000, 1000–2000, 2000–3000, 3000–4000, 4000–5000, 5000–6000, 6000–7000, 7000–8000, >8000 |
| DTRI (m) | 10 | <1000, 1000–2000, 2000–3000, 3000–4000, 4000–5000, 5000–6000, 6000–7000, 7000–8000, 8000–9000, >9000 |
| Aspect | 9 | Flat (-1), north (0–22.5, 337.5–360), northeast (22.5–67.5), east (67.5–112.5), southeast (112.5–157.5), south (157.5–202.5), southwest (202.5–247.5), west (247.5–292.5), northwest (292.5–337.5) |
| Lithology | 20 | $\in_w$, $\beta_\mu$, $T_3$, $T_2$, $S_?$, Q, $P_{Z1}$, $P_t$, $P_?$, $P_2$, $P_1$, $N_2$, $N_1$, $K_2$, $K_1$, $J_3$, $J_2$, $J_1$, $E_2$, $C_3$ |

**3.1.1 Information gain ratio.** The IGR is an improvement on the information gain method. Information gain indicates the degree of uncertainty reduction when certain factors are applied to judge landslide susceptibility in a certain area; it is the ratio of information gain to empirical entropy. The IGR value can indicate the importance of a factor: the higher the IGR value is, the more important the information provided by the factor for landslide susceptibility prediction [53]. The process for calculating IGR is as follows. $N$ is the landslide training dataset, and $|N|$ is the total number of sample points. $C_1$ and $C_2$ represent the numbers of landslide points and non-landslide points, respectively. Let $k = 1$ and 2; then, $\sum_{k=1}^{k} |C_k| = |N|$. There are $m$ factors ($A$) that affect the landslide, and the values of those factors are ($a_1, a_2, \ldots, a_m$). Factor $A$ divides $N$ into $n$ subcategories ($N_1, N_2, \ldots, N_n$) according to the number of categories. The sample points belonging to $C_k$ in $N_n$ are denoted as $N_{ik}$. The process for calculating IGR is as follows:

1. The empirical entropy $H(N)$ of the landslide training dataset $N$ is calculated:

$$H(N) = -\sum_{k=1}^{k} \frac{|C_k|}{|N|} \log_2 \frac{|C_k|}{|N|} \tag{1}$$

2. The empirical conditional entropy $H(N|A)$ of factor $A$ in the landslide training dataset $N$ is calculated:

$$H(N|A) = \sum_{i=1}^{n} \frac{N_i}{N} H(N_i) = -\sum_{i=1}^{n} \frac{N_i}{N} \sum_{k=1}^{k} \frac{N_{ik}}{N_i} \log_2 \frac{N_{ik}}{N_i} \tag{2}$$

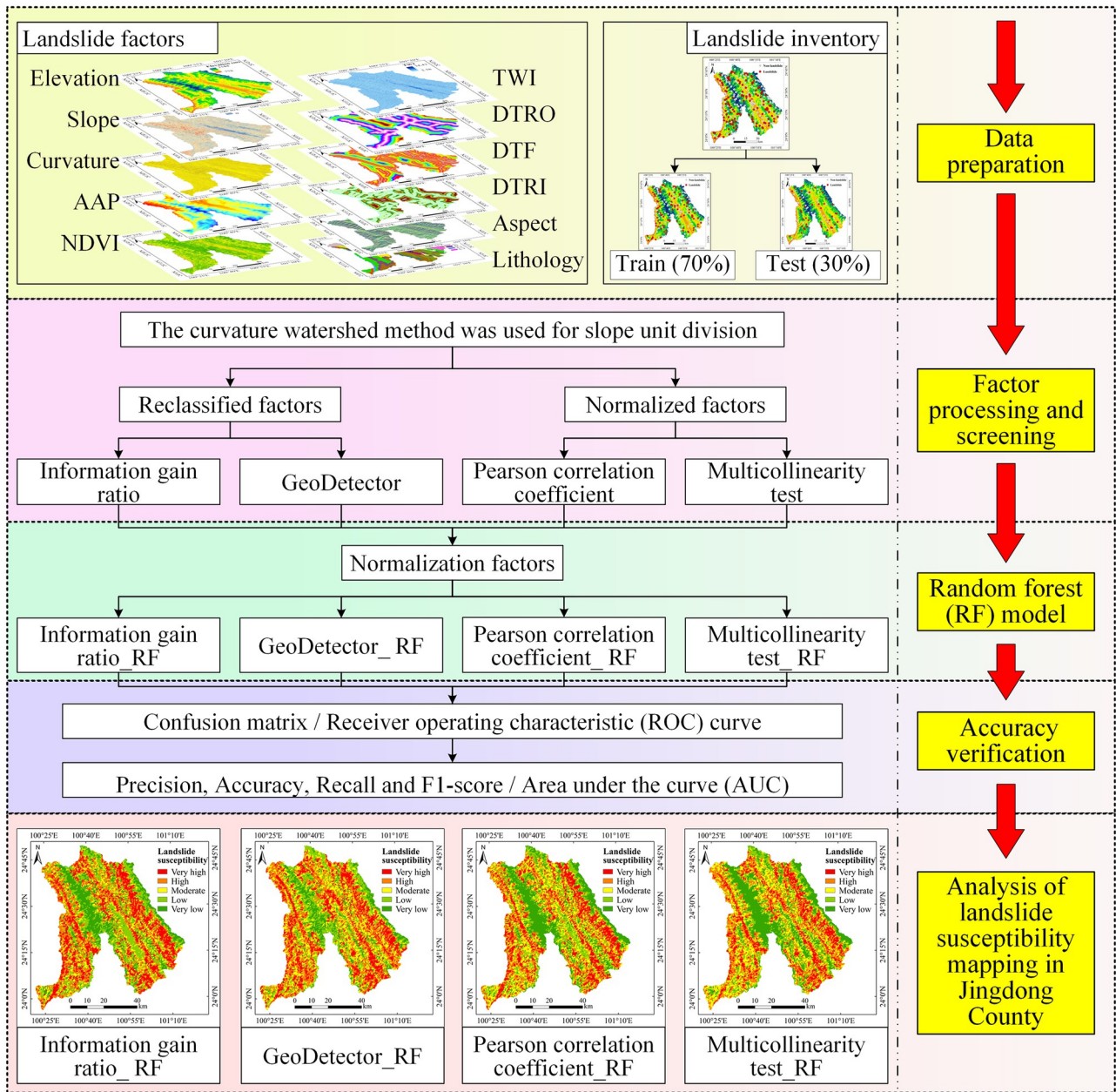

**Fig 3. Flowchart of the methods used in this study.**

3. The information gain *IG* of factor *A* is calculated:

$$IG(N, A) = H(N) - H(D|A) \tag{3}$$

4. The *IGR* of factor *A* is calculated:

$$IGR(N, A) = \frac{IG(N, A)}{H(N)} \tag{4}$$

**3.1.2 GeoDetector.** GD is a statistical method used to detect spatial differentiation and to reveal the driving factors behind it [54]. The core idea is that if a factor is assumed to have a significant influence on the occurrence of landslides, then the spatial distribution of that factor and landslides should be similar. This similarity is expressed by the ratio of the local variance to the global variance. In this paper, the factor detector included in GD was used to calculate the weight of each factor (GD software was obtained from http://www.geodetector.cn/). The factor detector was used to detect the extent to which a factor ($X$) explained the spatial pattern of a landslide ($Y$), which was measured by the $q$ value. The stronger the explanatory power was, the more important the factor. The equation for $q$ is:

$$q = 1 - \frac{1}{N\sigma^2} \sum_{h=1}^{L} N_h \sigma_h^2 \tag{5}$$

In Eq (5), $h$ = 1, 2. . .; $L$ is the number of subcategories of each factor; and $N_h$ and $N$ are the numbers of units in the $h$-th category and the entire study area for the factor, respectively. $\sigma_h^2$ and $\sigma^2$ are the variance in the $Y$ value of the $h$-th category and the entire study area for the factor, respectively. The value of $q$ ranges from 0 to 1. The larger the value of $q$ is, the stronger the explanatory power of the factor and the more important it is in model prediction.

**3.1.3 Pearson correlation coefficient.** PCC is an indicator that reflects the degree of correlation between two factors and is the product of the covariance of the two factors divided by the standard deviation. The values of the correlation coefficient range between -1 and 1. The greater the absolute value of the correlation coefficient is, the stronger the correlation between two factors. If the absolute value of the correlation coefficient between the two factors exceeds 0.7, then there is a high degree of correlation between the two factors [55]. The PCC calculation equation is:

$$r_{XY} = \frac{\sum_{i=1}^{n}(X_i - \bar{X})(Y_i - \bar{Y})}{\sqrt{\sum_{i=1}^{n}(X_i - \bar{X})^2}\sqrt{\sum_{i=1}^{n}(Y_i - \bar{Y})^2}} \tag{6}$$

In Eq (6), $X$ and $Y$ represent two landslide factors, $r_{XY}$ is the correlation coefficient between factor $X$ and factor $Y$, $n$ is the number of input training data, $X_i$ and $Y_i$ are the values of the $i$-th training data in $X$ and $Y$, respectively, and $\bar{X}$ and $\bar{Y}$ are the averages of $X$ and $Y$, respectively.

**3.1.4 Multicollinearity test.** Multicollinearity refers to the high degree of correlation among factors, which can result in biased landslide predictions. Reducing the correlation between factors is of great significance for improving the accuracy of landslide prediction. The variance inflation factor (VIF) and tolerance (TOL) are important indicators for judging whether there are multiple collinearities between factors, and these two indicators are the reciprocal of each other. In general, $VIF<10$ and $TOL>0.1$ indicate that there is no multicollinearity between factors; $VIF>10$ and $TOL<0.1$ indicate that there is multicollinearity between factors [56]. The equation for calculating the $VIF$ is:

$$VIF_i = \frac{1}{TOL} = \frac{1}{1 - R_i^2} \tag{7}$$

In Eq (7), $VIF_i$ is the variance expansion coefficient of the $i$-th factor, and $Ri$ is the negative correlation coefficient of the regression analysis of the $i$-th factor on the remaining factors. The value of $1 - R_i^2$ is the TOL of the $i$-th factor.

## 3.2. Landslide susceptibility mapping based on the random forest model

The RF model is a type of bagging ensemble algorithm. The core idea of the bagging algorithm is to construct multiple mutually independent classifiers and then, based on the prediction result of each classifier, to arrive at the final prediction result by averaging or voting. In solving a classification problem such as those presented by landslide data, the result is determined by voting, which is applied to determine whether a landslide will occur in a study area [57]. The specific steps for the construction of the RF model are as follows:

1. *N* extractions with replacement are performed for the original landslide training dataset (*N*) to form a new landslide training dataset (the number of samples in the new landslide training dataset is the same as that in the original landslide training dataset). The unextracted landslide sample points form the out-of-bag (OOB) dataset, which is used to evaluate the performance of the model. This process is repeated *k* times to obtain a total of *k* landslide training datasets and *k* corresponding OOB datasets.

2. Each landslide training dataset is used to generate a decision tree, generating a total of *k* decision trees. Assuming that each landslide point has *M* factors, *m* factors are randomly selected from the *M* factors ($m \leq M$). When each node of the decision tree is split, a certain method (such as the Gini criteria) is applied to the *m* factors to select a factor as the optimal splitting attribute of the node. Then, the Gini criterion at node k is as follows [58, 59]:

$$Gini_k = 1 - \sum_{i=1}^{n_{kc}} p_{ki}^2 \qquad (8)$$

In Eq (8), $n_{kc}$ denotes the number of classes in the set under consideration, and $p_{ki}$ denotes the proportion of the current class *i* in that set.

3. In the process of decision tree generation, the method in (2) is applied to each node until it cannot be split further (if the factor selected by the next node is the factor used when the parent node was split, the splitting stops). There is no pruning during the decision tree generation process.

4. The generated *k* decision trees are combined into an RF model, and voting is used to predict the probability of landslide susceptibility in the study area.

## 3.3. Accuracy verification

Accuracy validation is the process of assessing the accuracy and reliability of a model. Confusion matrices and receiver operating characteristic (ROC) curves are widely used in LSM for accuracy verification [60, 61]. The ROC curve is based on a confusion matrix that calculates the true positive rate (TPR) and the false positive rate (FPR) for different threshold scenarios. The curve was plotted with the FPR value as the abscissa and the TPR value as the ordinate. The area under the curve (AUC) of the ROC curve is a quantitative expression that can be applied to measure the effectiveness of model classification. The larger the AUC value is, the better the classification effect of the model [62]. The confusion matrix is a visual representation of a model's classification results against the actual categories and can help us better understand the classification performance of the model. For precision, accuracy, recall and F1 score

as evaluation metrics, the equations are as follows:

$$Precision = \frac{TP}{TP + FP} \tag{9}$$

$$Accuracy = \frac{TP + TN}{TP + FP + TN + FN} \tag{10}$$

$$Recall = \frac{TP}{TP + FN} \tag{11}$$

$$F1\ score\ = 2 \times \frac{Precision \times Recall}{Precision + Recall} \tag{12}$$

In Eqs (9)–(12), *TP* is the number of landslide points that are correctly classified as landslide points, *FN* is the number of landslide points that are incorrectly classified as non-landslide points, *TN* is the number of non-landslide points that are correctly classified as non-landslide points, and *FP* is the number of non-landslide points that are incorrectly classified as landslide points.

## 4. Results

### 4.1. Results of the four factor screening methods

In this study, a total of 11 initial factors were selected through preanalysis and screening. Then, the input factors were screened using different factor screening methods. The results of the four factor screening methods are shown below.

The results of the IGR are shown in Fig 4. The results show that NDVI has the greatest impact on the model's predictive ability (0.148), followed by elevation (0.074), aspect (0.054), DTF (0.046), DTRO (0.045), TWI (0.039), rainfall (0.035), slope (0.031)), DTRI (0.020) and

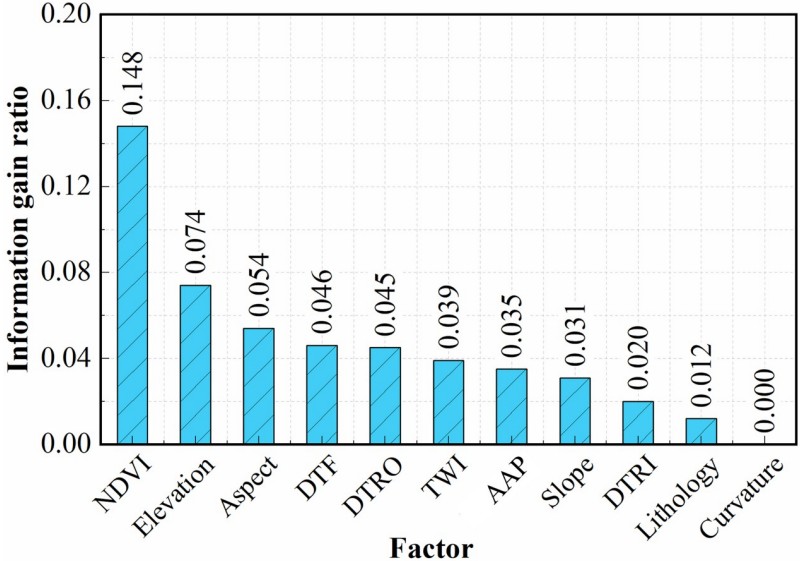

**Fig 4. Results of the information gain ratio (IGR) analysis of the landslide factors.**

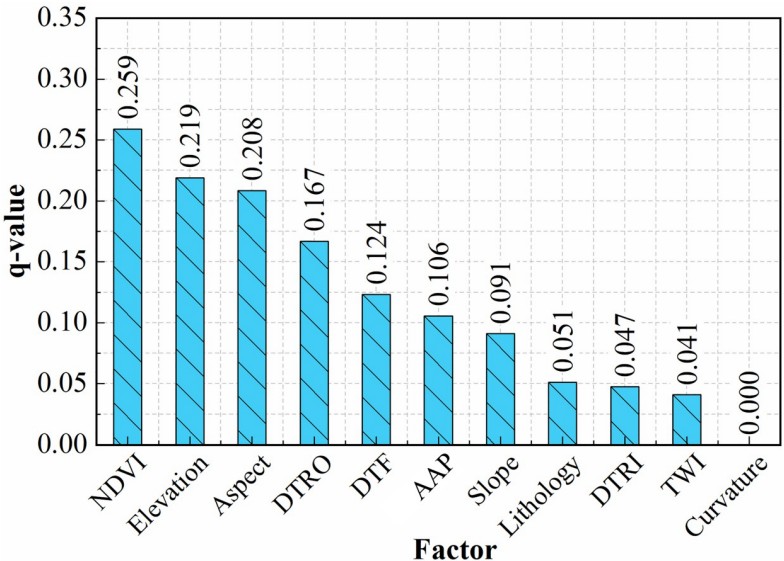

**Fig 5. The results of the GeoDetector (GD) analysis of the landslide factors.**

lithology (0.012). The last factor, curvature, has an IGR value of 0, indicating that this factor does not provide useful information for model prediction. Therefore, the 10 factors with IGR values greater than 0 were selected from the 11 initial factors for use in the RF model [63], and 1 factor was eliminated.

The GD results are shown in Fig 5. The results show that NDVI has the greatest impact on the model's predictive ability (0.259), followed by elevation (0.219), aspect (0.208), DTRO (0.167), DTF (0.124), rainfall (0.106), slope (0.091), lithology (0.051)), DTRI (0.047), TWI (0.041) and curvature (0). Therefore, the 8 factors with $q$ values greater than 0.05 were selected from the 11 initial factors for use in the RF model [64], and 3 factors were eliminated.

The results of the PCC are shown in Fig 6. Factors with absolute correlation coefficients exceeding 0.7 were identified [65, 66]. Slope and TWI were highly correlated (-0.77). Slope is an important physical condition for landslide occurrence, so the TWI was eliminated. Therefore, 10 factors with absolute correlation coefficients less than 0.7 were selected from the 11 initial factors for the RF model, and 1 factor was eliminated.

The results of the MTs, VIF and TOL are shown in Fig 7. All the factors met the factor screening criteria, with $VIF<10$ and $TOL>0.1$, and there was no multicollinearity among the factors. Therefore, all 11 factors were introduced into the RF model, and no factors were eliminated. Additionally, this model was used as a comparison model without factor screening.

## 4.2. Accuracy verification

The factors screened by the different factor screening methods were substituted into the RF model for landslide susceptibility modeling. The statistical results using the precision, accuracy, recall and F1 score are presented in Table 3. Among the precision indicators, IGR_RF and GD_RF showed the best results. In the accuracy metric, IGR_RF, GD_RF and PCC_RF showed the best results. In the recall and F1 metrics, PCC_RF results performed best. These results indicate that the predictive power of the models using factor screening (IGR_RF, GD_RF and PCC_RF) is improved compared to the results without elimination of factors (MT_RF), demonstrating the effectiveness of factor screening. In addition, ROC curves were

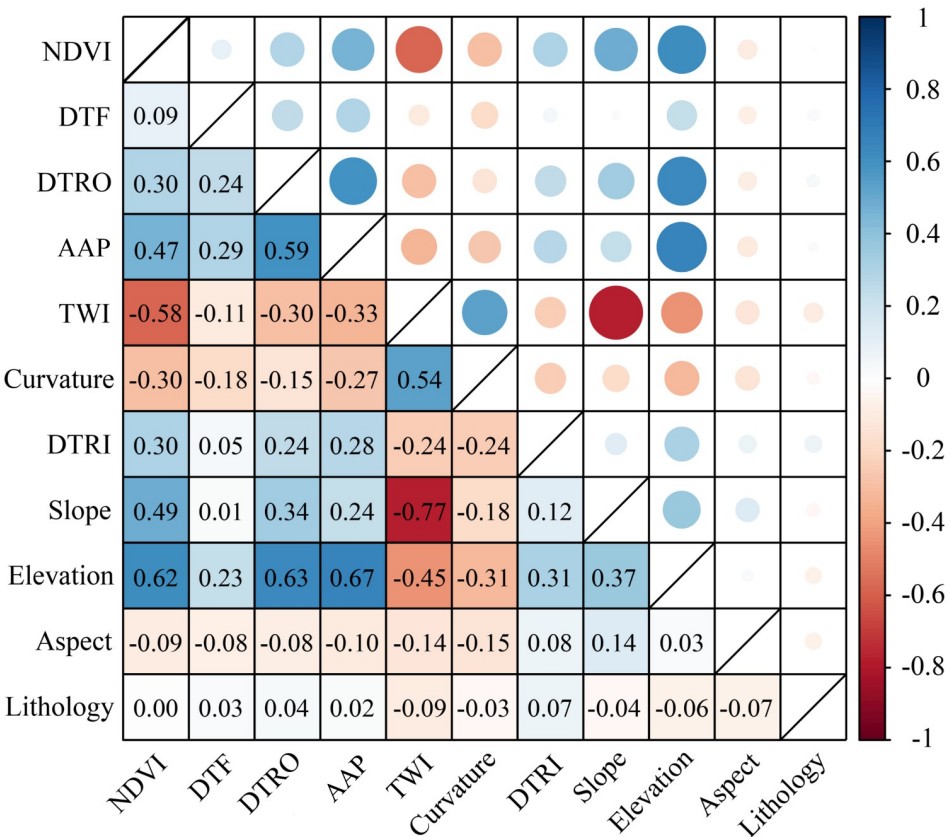

**Fig 6. The Pearson correlation coefficients (PCCs) of the landslide factors.**

used to assess the predictive power of the models (Fig 8). The predictive power of all four models was above 0.9, indicating that the RF models have good predictive performance. The IGR_RF model had the largest AUC value (0.9334), followed by GD_RF (0.9256), PCC_RF (0.9197), and MT_RF (0.9194).

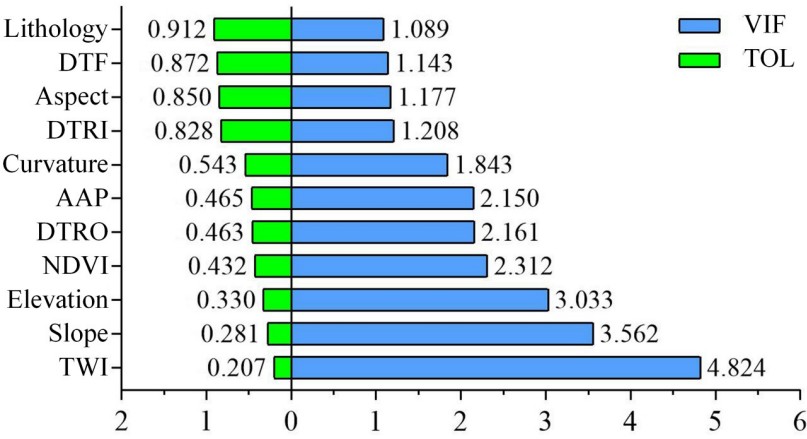

**Fig 7. The results of the multicollinearity tests (MTs) of the landslide factors.**

**Table 3. Performance of different factor screening methods.**

| Method | Precision | Accuracy | Recall | F1 score |
|---|---|---|---|---|
| IGR_RF | 0.8974 | 0.8780 | 0.8537 | 0.8750 |
| GD_RF | 0.8974 | 0.8780 | 0.8537 | 0.8750 |
| PCC_RF | 0.8780 | 0.8780 | 0.8780 | 0.8780 |
| MT_RF | 0.8537 | 0.8537 | 0.8537 | 0.8537 |

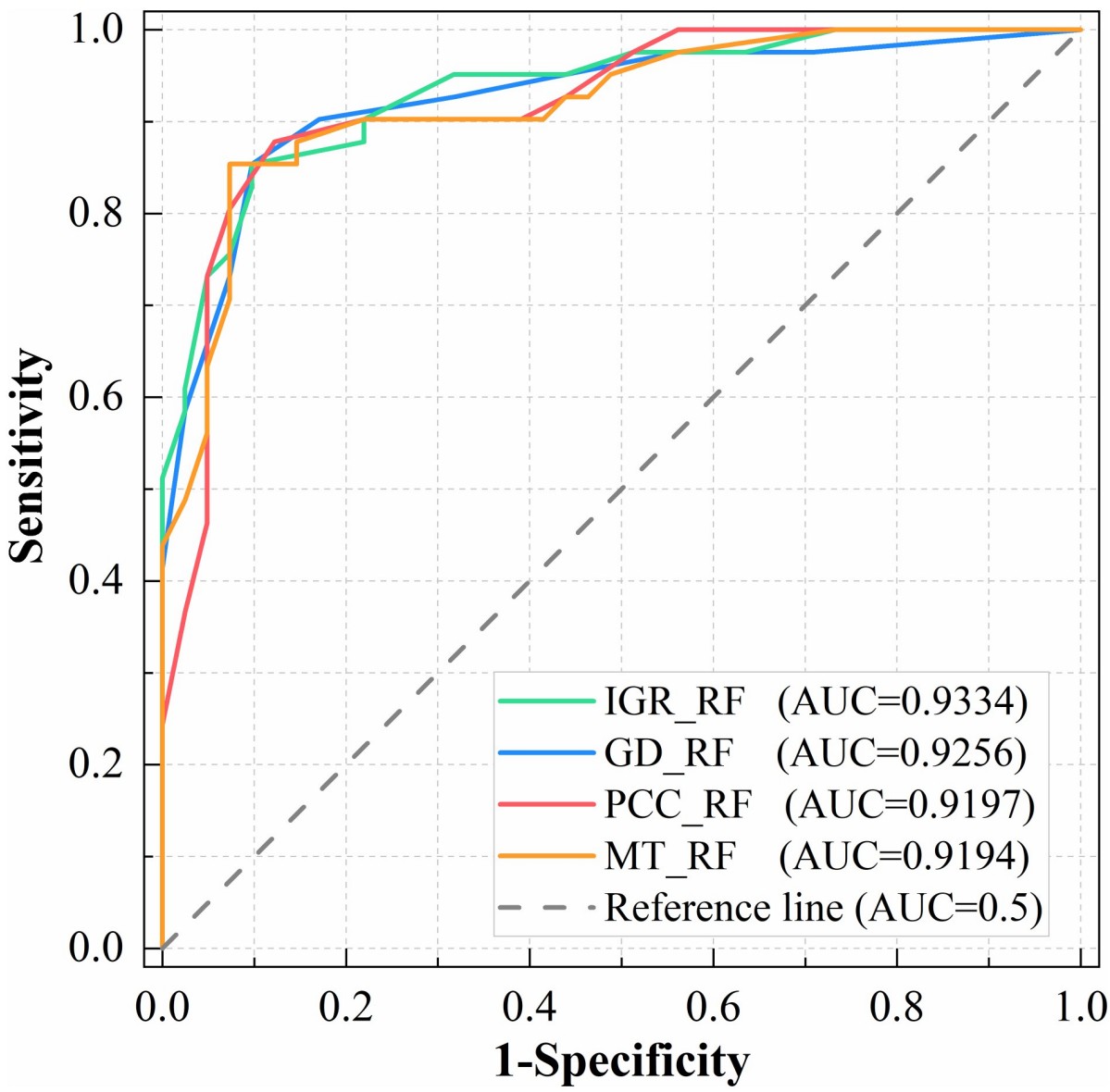

**Fig 8. The ROC curve results obtained using four factor screening methods.**

## 4.3. Landslide susceptibility mapping

The RF model classifiers trained with four factor screening methods were applied to create probability predictions for the whole study area, and four probability maps of landslide susceptibility in Jingdong County were generated. We used the natural breaks (Jenks) method to divide the study area into five levels of susceptibility: very low, low, moderate, high and very high [67]. The results of this classification are shown in Fig 9. The spatial distribution patterns of the four predicted outcomes are similar, with very high and very low susceptibility zones distributed in strips due to topography and human activity. The results of the zoning statistics

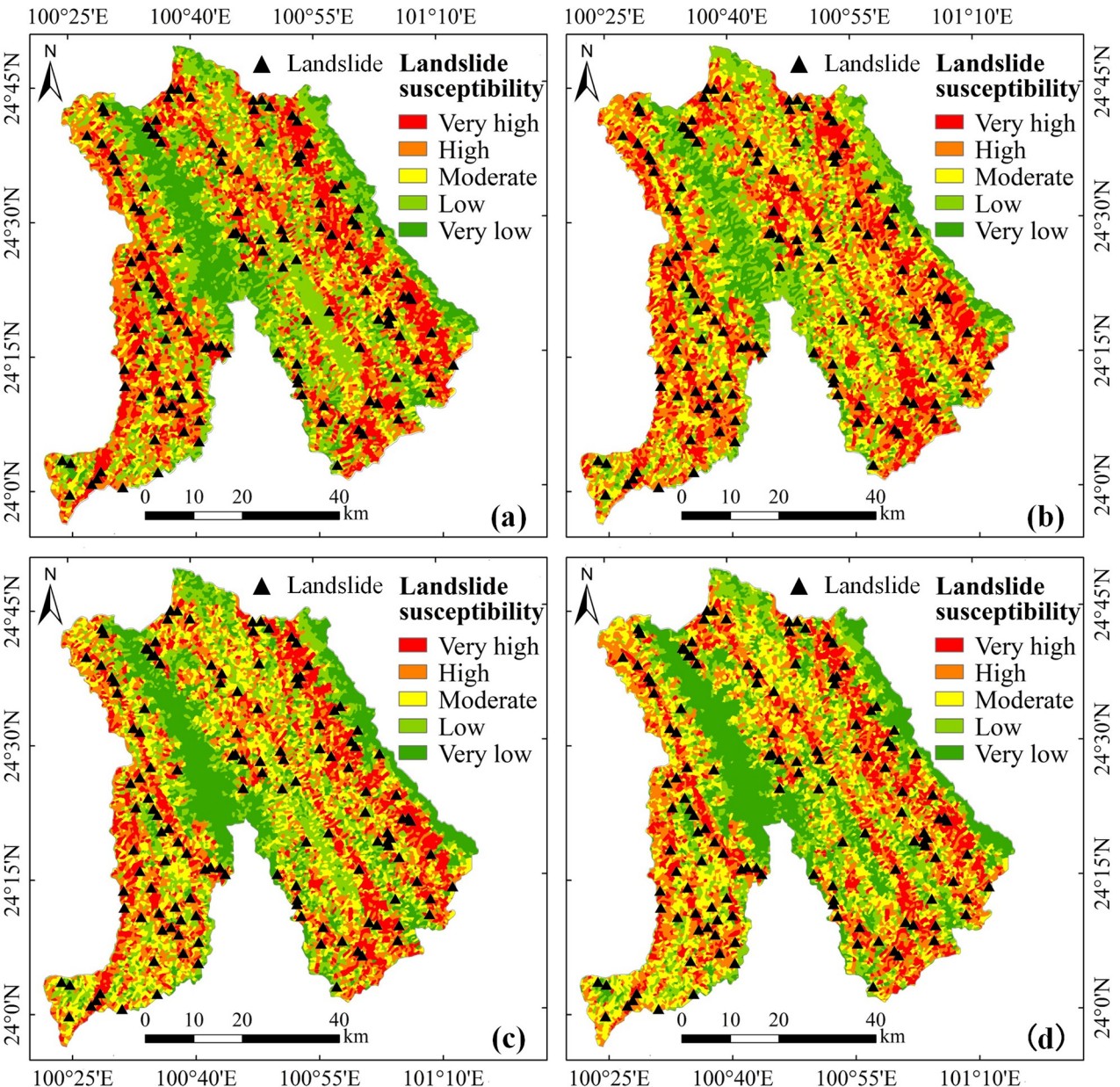

**Fig 9.** The results of landslide susceptibility mapping based on four different factor screening methods: (a) IGR_RF; (b) GD_RF; (c) PCC_RF; (d) MT_RF.

**Table 4. Statistical results of landslide susceptibility classification of four models.**

| Model | Susceptibility class | Landslide (point) | Landslide ratio | Area (km$^2$) | Area ratio | Frequency ratio |
|-------|---------------------|-------------------|-----------------|---------------|------------|-----------------|
| IGR_RF | Very low | 1 | 2.44% | 648.79 | 14.53% | 0.168 |
| | Low | 2 | 4.88% | 1074.58 | 24.06% | 0.203 |
| | Moderate | 4 | 9.76% | 669.36 | 14.99% | 0.651 |
| | High | 13 | 31.71% | 1068.76 | 23.93% | 1.325 |
| | Very high | 21 | 51.22% | 1004.80 | 22.50% | 2.277 |
| GD_RF | Very low | 1 | 2.44% | 419.81 | 9.40% | 0.259 |
| | Low | 1 | 2.44% | 889.78 | 19.92% | 0.122 |
| | Moderate | 4 | 9.76% | 973.34 | 21.79% | 0.448 |
| | High | 18 | 43.90% | 1214.94 | 27.20% | 1.614 |
| | Very high | 17 | 41.46% | 968.41 | 21.68% | 1.912 |
| PCC_RF | Very low | 0 | 0.00% | 797.40 | 17.85% | 0.000 |
| | Low | 4 | 9.76% | 884.01 | 19.79% | 0.493 |
| | Moderate | 7 | 17.07% | 979.74 | 21.94% | 0.778 |
| | High | 15 | 36.59% | 969.21 | 21.70% | 1.686 |
| | Very high | 15 | 36.59% | 835.93 | 18.72% | 1.955 |
| MT_RF | Very low | 3 | 7.32% | 903.60 | 20.23% | 0.362 |
| | Low | 1 | 2.44% | 736.08 | 16.48% | 0.148 |
| | Moderate | 5 | 12.20% | 1038.65 | 23.26% | 0.524 |
| | High | 16 | 39.02% | 1056.79 | 23.66% | 1.649 |
| | Very high | 16 | 39.02% | 731.16 | 16.37% | 2.384 |

are shown in Table 4. The frequency ratio is the ratio of the landslide point ratio to the area ratio within each classification type. Within the very high and high susceptibility areas, all four models predicted more than 70% of the historical landslides, with the GEO_RF model predicting the greatest number (85.37%). In contrast, the MT_RF model without elimination of factors misidentified the highest number of landslides (3 landslides) in the very low susceptibility area. This shows that redundant factors can provide the model with incorrect information and cause misjudgments. In general, the frequency ratio increases gradually as the susceptibility level rises, with all four models showing an upward trend. The MT_RF model has the highest frequency ratio (2.384) in the very high susceptibility area but has only 16 landslides in the very high susceptibility area. The IGR_RF model has the second highest frequency ratio (2.277) after MT_RF in the very high susceptibility area and predicted 21 historical landslides in the very high susceptibility area. Clearly, factor screening facilitates the improvement of prediction accuracy.

## 5. Discussion

### 5.1. Importance of factor screening

Landslides are influenced by a variety of factors, such as topography, geomorphology, geological structure, hydrology, meteorology and human activities. Geography varies from region to region, and the magnitude of the influence of the factors on landslide formation may also vary [68]. How to quantitatively select the factors that are closely related to landslide formation and improve the accuracy and reliability of the model is an important issue. Factor screening is based on the criterion of correlation between factors or importance to the landslide, optimizing the quality of the input data, removing redundant or noisy information and improving the accuracy of the LSM.

Four factor screening methods (IGR, GD, PCC and MT) were selected for comparative analysis. The number of factors eliminated by IGR, GD and PCC was one, three and one, respectively. In contrast, the results of the MT concluded that all factors met the requirements, indicating that the MT model is insensitive to correlations between factors, in keeping with the previous screening results [69, 70]. We used the MT model as the original model without elimination of factors and used it as a comparison for the remaining three models. This analysis found that the predictive power of the model implementing factor screening was significantly improved in LSM (Fig 8, Tables 3 and 4). The AUC values for IGR_RF, GD_RF, and PCC_RF were 0.9334, 0.9256, and 0.9197, respectively, compared to 0.9194 for the original model without elimination. IGR_RF predicted five more landslides in the very high susceptibility zone than the original model. Moreover, within the very low susceptibility area, the original model predicted three landslides, while IGR_RF, GD_RF and PCC_RF predicted one, one and zero, respectively. It can thus be seen that using more factors does not necessarily improve the prediction accuracy of the model, and redundant factors provide incorrect information to the prediction model. In contrast, using the right factors can lead to better prediction results. Furthermore, given the same number of eliminated factors (IGR and PCC models), IGR obtained better prediction results because PCC analyses the correlation and redundancy between factors and does not consider the relationship between factor attribute characteristics and landslides. The IGR, on the other hand, is based on the probability of landslides occurring under different factor conditions and accounts for the relationship between factor attribute characteristics and landslides. Overall, the IGR_RF model yields better prediction results.

## 5.2. The influence of screened factors on landslide susceptibility

Based on the historical landslide data and the selected factor data, both IGR and GD methods can rank the weights of the selected factors. The top three factors screened by both methods are NDVI, elevation, and aspect (Figs 4 and 5). Analysis of the relationship between landslides and factors through landslide point density is shown in Fig 10. As shown in Fig 10(a), the density of landslide occurrence increases and then decreases with increasing NDVI values. Areas with NDVI values of 0.4–0.6 are most susceptible to landslides. Low NDVI values are mainly concentrated in the plains of river valleys where human activities are taking place. These areas are relatively flat and are not conducive to landslide formation. Areas with high NDVI values are mainly in the two nature reserves of Wuliang Mountain and Ailao Mountain, which still maintain intact and pristine ecology with high vegetation cover. High vegetation cover can effectively reduce the occurrence of landslide hazards, especially shallow landslides [71]. Fig 10 (b) reflects the relationship between the DEM data and landslides. With increasing elevation, the landslide occurrence density first increases and then decreases. This is consistent with the results of studies in some other areas of Yunnan Province [72, 73]. Landslides occur between 950 m and 2200 m, where human activity is concentrated. Human development and modification tend to change geological and hydrological conditions, making slopes less stable and more prone to landslides. Landslide events cease to occur when the elevation is greater than 2200 m. These areas are mainly in the Wuliang Mountain and Ailao Mountain Nature Reserve, where the elevations are relatively high, human activity is low, and the root systems of vegetation enhance the soil's resistance to landslides. Fig 10(c) reflects the relationship between different aspects and landslides. Landslides occur mainly in west-, north- and northwest-oriented areas and less commonly in the east-oriented areas. This is mainly due to the influence of the northwest-southeast-trending mountain ranges in the region, which block the southwest warm and humid air currents as they advance, resulting in more precipitation in the west than in the east and higher temperatures in the west than in the east at the same latitude

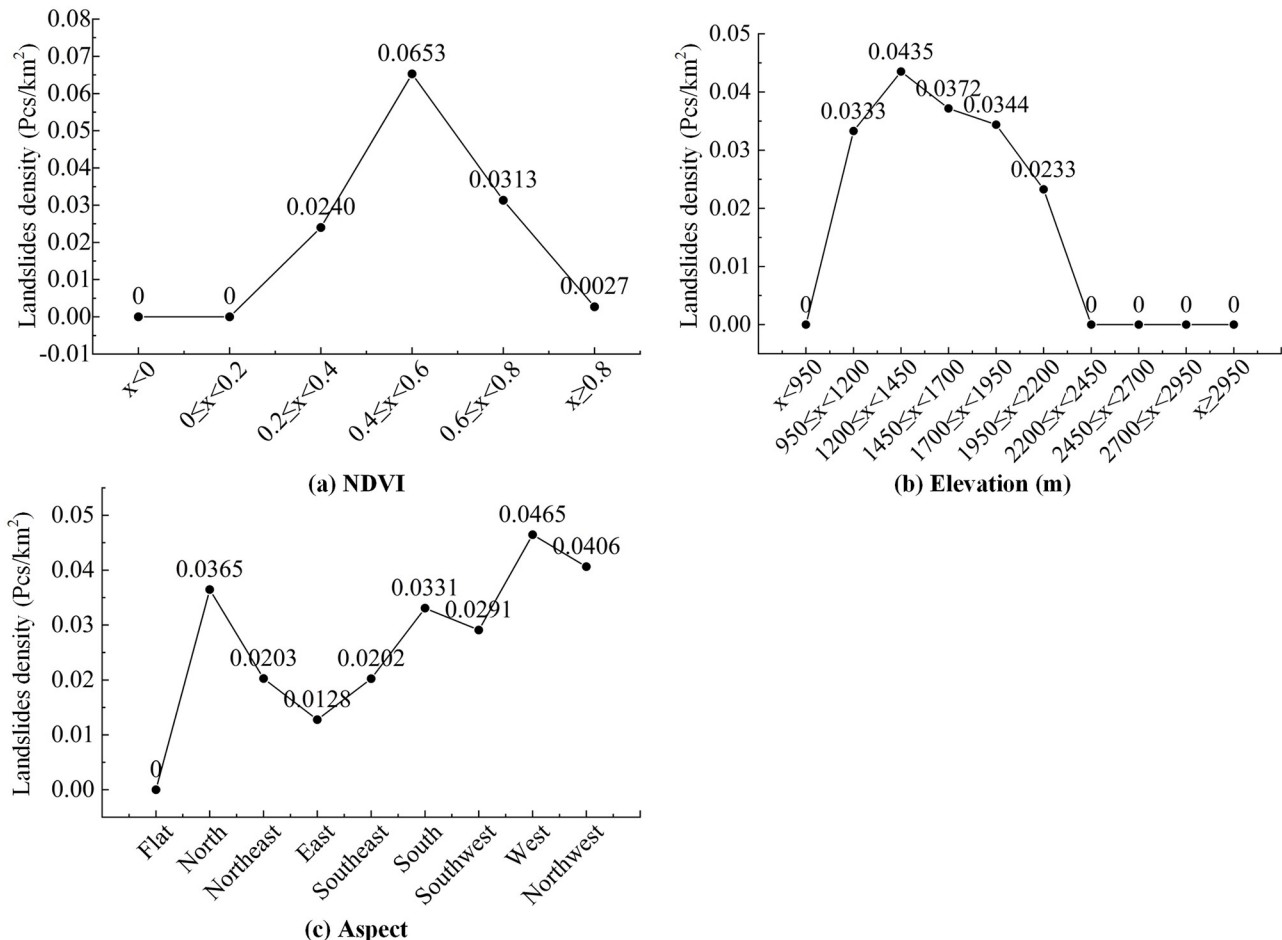

**Fig 10.** Density statistical analysis of the main factors: (a) NDVI, (b) elevation, and (c) aspect.

and elevation. Due to the influence of rainfall and temperature, landslides are prone to occur on west-facing slopes.

## 5.3. Uncertainty and limitations of this study

The importance of factor screening has been demonstrated through the use of different factor screening methods, but there are still some uncertainties and limitations in this study. First, factor data can vary in format and resolution depending on the source, and this issue is common in LSM [74–76]. Most of the factors in the paper are based on data with a resolution of 30 m. The average annual rainfall data are 1,000 m resolution data interpolated based on observations from meteorological stations. The fault and lithology data were derived from 1:200,000 geological survey data. In combination with data availability and later calculations, we resampled all factors to a 30 m × 30 m resolution raster data format and unified them under the same projection coordinate system. From the predicted results, it is clear that use of the resampling method is feasible. In future landslide surveys and data collection, we will consider acquiring more detailed rainfall and geological data to improve the quality of factor data.

Second, there is the problem of dividing slope units. We have used the curvature watershed method, which is relatively effective in classifying slope units. This method is based on the

topography of the terrain, and the area of the slope unit is relatively uniform with little internal topographic variation. In recent years, with the development of remote sensing technology, interferometric synthetic aperture radar technology has been increasingly applied in the field of surface deformation monitoring and can quantitatively describe surface deformation [77]. In future research, we will refer to deformation variables to classify slope units to improve the quality of slope classification.

Finally, the RF model was chosen for the LSM, and good results were obtained. In future research, we will use different machine learning models or ensemble models, as well as optimization of hyperparameters in the model, to improve the predictive performance of the model.

## 6. Conclusions

For machine learning models, the quality of the input factor data affects the prediction accuracy of the model. Factor screening plays an important role as the starting point for data input. To compare and analyze the effects of different factor screening methods on the prediction results, four factor screening methods—IGR, GD, PCC and MT—were used in this study. The 2014 landslide hazard data from Jingdong County, Yunnan Province, were used as an example, and the slope unit was used as the basic evaluation unit. Each of the four factor screening methods was combined with the RF model for susceptibility assessment. The results of the study showed the following:

1. Compared with the prediction results without eliminating factors, factor screening is beneficial to improving the prediction accuracy of the model, and there are some differences in the prediction results among the different screening methods. IGR and GD consider the relationships between factors and landslides and can calculate factor weights. PCC and MT do not calculate the weights of factors.

2. The results of the ROC curve analysis show that the IGR_RF model has a higher AUC value (0.9334) than the other models (GD_RF, PCC_RF, MT_RF). The IGR_RF model predicted the most landslides in the very high susceptibility area.

3. Based on the results of the factor screening with the IGR and GD methods, the top 3 factors are NDVI, elevation and aspect. Curvature has the least influence on landslides. The influence of the main factors on landslides was analyzed in relation to the actual situation in Jingdong County. Landslides mainly occur in areas with NDVI values of 0.4–0.6, elevations of 950 m-2200 m and a west-facing orientation.

In summary, IGR_RF predicts the best results. IGR is calculated based on the probability of landslides occurring under different factor conditions, which considers the relationship between factor attribute characteristics and landslides and improves the reliability of the input data. In addition, the method can give the weight values of different factors to provide a reference for landslide prevention and management. It provides a reference for other scholars in factor screening.

## Acknowledgments

The author would like to express thanks to anonymous reviewers for all careful review of the paper and kind suggestions made to improve overall quality of the manuscript.

## Author Contributions

**Conceptualization:** Tengfei Gu, Jia Li, Ping Duan.

**Formal analysis:** Tengfei Gu.

**Funding acquisition:** Jia Li, Ping Duan.

**Methodology:** Tengfei Gu, Jia Li.

**Resources:** Mingguo Wang, Ping Duan.

**Software:** Tengfei Gu.

**Supervision:** Jia Li.

**Validation:** Mingguo Wang, Ping Duan.

**Visualization:** Libo Cheng.

**Writing – original draft:** Tengfei Gu.

**Writing – review & editing:** Tengfei Gu, Jia Li, Ping Duan, Yanke Zhang.

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
