## [Decision Letter · Decision Letter 0]

18 May 2023

PONE-D-23-12592Study on landslide susceptibility mapping with different factor screening methods and random forest modelsPLOS ONE

Dear Dr. Li,

Thank you for submitting your manuscript to PLOS ONE. After careful consideration, we feel that it has merit but does not fully meet PLOS ONE’s publication criteria as it currently stands. Therefore, we invite you to submit a revised version of the manuscript that addresses the points raised during the review process. Please submit your revised manuscript by Jul 02 2023 11:59PM. If you will need more time than this to complete your revisions, please reply to this message or contact the journal office at plosone@plos.org. Please include the following items when submitting your revised manuscript:A rebuttal letter that responds to each point raised by the academic editor and reviewer(s). You should upload this letter as a separate file labeled 'Response to Reviewers'.A marked-up copy of your manuscript that highlights changes made to the original version. You should upload this as a separate file labeled 'Revised Manuscript with Track Changes'.An unmarked version of your revised paper without tracked changes. You should upload this as a separate file labeled 'Manuscript'.

We look forward to receiving your revised manuscript.

Kind regards,

Salim Heddam

Academic Editor

PLOS ONE

Journal Requirements:

3. We note that Figures 1,2 and 3 in your submission contain map/satellite images which may be copyrighted. All PLOS content is published under the Creative Commons Attribution License (CC BY 4.0), which means that the manuscript, images, and Supporting Information files will be freely available online, and any third party is permitted to access, download, copy, distribute, and use these materials in any way, even commercially, with proper attribution. For these reasons, we cannot publish previously copyrighted maps or satellite images created using proprietary data, such as Google software (Google Maps, Street View, and Earth). For more information, see our copyright guidelines: http://journals.plos.org/plosone/s/licenses-and-copyright.

a. You may seek permission from the original copyright holder of Figures 1,2 and 3 to publish the content specifically under the CC BY 4.0 license.  

Reviewers' comments:

Reviewer's Responses to Questions

**Comments to the Author**

1. Is the manuscript technically sound, and do the data support the conclusions?

Reviewer #1: Yes

Reviewer #2: Partly

Reviewer #3: Yes

2. Has the statistical analysis been performed appropriately and rigorously? 

Reviewer #1: I Don't Know

Reviewer #2: No

Reviewer #3: No

3. Have the authors made all data underlying the findings in their manuscript fully available?

Reviewer #1: Yes

Reviewer #2: No

Reviewer #3: No

4. Is the manuscript presented in an intelligible fashion and written in standard English?

Reviewer #1: No

Reviewer #2: Yes

Reviewer #3: Yes

5. Review Comments to the Author

Reviewer #1: Dear Authors,

This paper explores the study on landslide susceptibility mapping with different factor screening methods and random forest models in Jingdong County, Yunnan Province, China. Most applied methods are not new but have been used before in several published studies. The subject is interesting. I want to encourage consider the more detailed comments below as major revisions and to improve the manuscript before publication in the PLOS ONE journal.

These are my main points of criticism, but on top of this, I find many other aspects of this paper problematic:

Abstract

1- Abstract section is so general and does not have a scientific structure as a research paper.

2- The obtained result has not been appropriately shown, and the authors should mention the obtained results of applied methods, not general notes, in the abstract section.

Introduction

In the first paragraph, the authors did not mention any references or statistics about landslides in the current study area and the rate of damages or losses of lives by this natural hazard.

The research background about the proposed subject and methods has not been appropriately mentioned, mainly applying approaches widely used in landslide susceptibility mapping in previous studies. Please update it with new references such as:

- https://doi.org/10.3390/f11080830

- https://doi.org/10.3390/f11040421

- https://doi.org/10.1016/j.catena.2018.01.012

Also, why did you not explain the novelty of the research adequately? Suppose authors utilized RF method implemented and assessed for LSM. Why did they not bring significant scientific resources in relation to the suggested model? Suppose the authors claim that the proposed methods are a new contribution to this research. In that case, they must bring defendable documents such as published scientific papers for their claim, clear it properly, and not bring the general notes found in any previously published papers.

The main goal of the paper should be better explained. It is not clearly stated. The goal should clearly and concisely explain the main scientific contributions of this work.

2.1.Study area

I did not see any description of the occurred landslides in the study area. Why do you choose this area for study? This section is a problem statement, and you should bring a logical reason for starting this research.

2.3. Factor data sources

How do you select non-landslide locations?

Figure 1: It is better separate landslide locations based on training and validation ratio.

Figure 2: It is better to add landslide points on all factors. The fuzzy classification is not acceptable for conditioning factors. It is better use different classification methods such as natural breaks, Quantile and etc. Furthermore, you should add the names of all factors in figure captions.

3. Methods

It is better to add applied equations for applied models by mathematical attitude for this section.

4. Results and Discussion

It is noted that a critical analysis of results has not been reported in the paper. The result section is so poor in writing and obtaining results. Also, the written results are not homogeneous due to no logical relationship between applied methods results.

Furthermore, I did not see any note about the comparison between the obtained results of the current research and previous research.

Figure 9: It is better to add landslide points on all parts of this figure.

Reviewer #2: I think there are doubts issues about this manuscript following:

1. I don't think the objectives and the rationale of the study clearly stated.

Combined some previous studies and based on my experience, the performance of landslide susceptibility model (LSM) is related to the types of Mapping units (grid / slope units, Doi:10.1016/j.gr.2022.07.013.), and depends on algorithm selection (Doi: 10.1016/j.enggeo.2020.105972.), algorithm parameter optimization (Doi: 10.1016/j.geomorph.2020.107201.), Selection of samples (especially negative samples, Doi: 10.1007/s12583-020-1072-9), selection of conditioning factors (Doi: 10.1016/j.geomorph.2021.107623, Doi: 10.1016/j.gsf.2021.101211.), and scale / size of mapping units (Doi: 10.1007/s10064-021-02415-y, Doi: 10.1016/j.catena.2022.106428.). The importance of factor screening for LSM performance should be highlighted in the Introduction section. Meanwhile, It is strongly recommended to highlight the contribution of this paper in the "Introduction," "Discussion," or "Conclusion" sections.

2. I don't think there are sufficient analysis about previous literature in the section Introduction. In particular, there is a lack of some relevant literature in recent years on the screening conditioning factors of landslide susceptibility. I strongly recommends to refer the following previous literature in the section Introduction, and further clearly state the research status of the factor screening for LSM.

Liao et al., (2022), https://doi.org/10.1016/j.catena.2022.106428

Wen et al., (2023), https://doi.org/10.1016/j.jenvman.2023.118177

Wen et al., (2022), https://doi.org/10.1080/10106049.2022.2120547

Sun et al., (2022), https://doi.org/10.1016/j.gr.2022.07.013

Zhou et al., (2022), https://doi.org/10.1080/10106049.2022.2076928

Zhang et al., (2021), https://doi.org/10.1007/s10064-021-02415-y

Sun et al., (2021), https://doi.org/10.3389/feart.2021.713803

Zhou et al., (2021), https://doi.org/10.1016/j.gsf.2021.101211

Sun et al., (2021), https://doi.org/10.1016/j.geomorph.2021.107623

3. Only 11 initial factors, which seems too few, and important conditioning factors could be missing for landslide susceptibility.

4. It is not comprehensive to only use AUC value to access the LSMs’ performance and comprise the effects of different LSMs. It should be added the confusion matrix and the corresponding accuracy, precision, recall and F1-score. For example, even if the AUC value is very high, if the recall rate of landslide is low, it indicates that the model is not a good one.

5. In fact, some researchers believe that factor screening is not important. It is strongly recommended to add RF-based LSM without screening factors and compare it with the LSMs’ performance under the 4 different factor screening methods.

Reviewer #3: The article employed IGR, GD, PCC, and MT methods for screening landslide conditioning factors and constrinfucted four landslide susceptibility models using the RF algorithm. The accuracy of these four methods was compared, and the research content of the article holds significance for landslide prevention and management. However, the article lacks substantial innovation, as the selected methods are commonly used in LSM. The discussion section is superficial, and certain sentences in the article are not coherent. It is recommended that the article undergo major revision and be reevaluated. Specific recommendations are provided below:

1. The author chose the RF algorithm because of its strong generalization ability and resistance to overfitting. However, based on my understanding, Boosting algorithms in ensemble learning, such as XGBoost and LightGBM, are better suited for addressing these issues.

2. In my opinion, Random Forest consists of a large number of decision trees. Even if a factor has a minimal impact on landslide development in a specific area, the model will still learn this phenomenon. Please provide a detailed explanation of how redundant factors affect the accuracy of the model.

3. It is suggested to include information about rainfall and vegetation coverage in the overview of the study area.

4. In Table 1, the article states that the grid resolution is 30m. The resolution of NDVI and elevation is also 30m, which is acceptable. However, the resolution of the rainfall data mentioned in the article is 1km. Inconsistent resolutions can significantly impact the accuracy of the model.

5. Why was reclassification performed? Additionally, most studies use the natural breakpoints (jenks) method for reclassification. Why did the author choose equal intervals for reclassification?

6. None of the four methods used by the author for screening were subjected to a significance test. Results without a significance test lack persuasiveness. Please include the results of the significance test.

7. In lines 292-296, what criteria were used to determine strong correlation? The correlation coefficient between elevation and rainfall is 0.63, indicating a strong correlation. Why wasn't it eliminated?

8. It is recommended to divide the results and discussions into two sections. The discussion section should delve deeper into the relationship between factors and landslide mechanisms.

9. Figure 8: The author only compared the accuracy of the four factor methods combined with the RF model, which is incomplete. To enhance the persuasiveness of the article, the author should include a comparative model - an RF model without factor screening.

10. It is advisable to present the conclusions in bullet points.

11. What specific aspect does the factor "Rainfall" refer to in the article? Is it the average annual rainfall or something else? Please clarify this point.

6. PLOS authors have the option to publish the peer review history of their article (what does this mean?). If published, this will include your full peer review and any attached files.

Reviewer #1: No

Reviewer #2: No

Reviewer #3: No

---

## [Author Response · Author response to Decision Letter 0]

10 Jul 2023

Dear editor:

Thank you for giving an opportunity to revise our manuscript, we appreciated editor and reviewers very much for their positive and constructive comments and suggestions on our manuscript entitled “Study on landslide susceptibility mapping with different factor screening methods and random forest models” (ID: PONE-D-23-12592R1).

We have studied the comments carefully and have made revision in this paper. Responses to comments from the journal and three reviewers can be found in the following sections. In addition, we have polished the language in the revised manuscript.

Sincerely,

Tengfei Gu

Journal Requirements: 

We have revised the formatting of the paper in line with the style requirements of PLOS ONE.

Data Availability Statement: Some of the original data used in this paper are confidential, including the landslide map and 1: 200,000-scale geological map, we can't share them publicly online. Landslide data from the Yunnan Bureau of Geology and Mineral Resources (http://www.yndkch.com), 1: 200,000-scale geological map from the National Geological Archives of China (http://www.ngac.org.cn/DataSpecial/geomap.html). Researchers who meet the criteria for access to confidential data may apply from the above institutions. All other data can be downloaded through the download link in the manuscript.

3. We note that Figures 1,2 and 3 in your submission contain map/satellite images which may be copyrighted. All PLOS content is published under the Creative Commons Attribution License (CC BY 4.0), which means that the manuscript, images, and Supporting Information files will be freely available online, and any third party is permitted to access, download, copy, distribute, and use these materials in any way, even commercially, with proper attribution. For these reasons, we cannot publish previously copyrighted maps or satellite images created using proprietary data, such as Google software (Google Maps, Street View, and Earth). For more information, see our copyright guidelines: http://journals.plos.org/plosone/s/licenses-and-copyright.

a. You may seek permission from the original copyright holder of Figures 1,2 and 3 to publish the content specifically under the CC BY 4.0 license. 

We have removed the satellite images in Figures 1, 2 and 3 and replaced them with other data.

Figures 1, 2 and 3: The base map of the elevation was downloaded from USGS National Map Viewer. The other data was produced by ArcGIS software.

Reviewer #1:

Dear Authors,

This paper explores the study on landslide susceptibility mapping with different factor screening methods and random forest models in Jingdong County, Yunnan Province, China. Most applied methods are not new but have been used before in several published studies. The subject is interesting. I want to encourage consider the more detailed comments below as major revisions and to improve the manuscript before publication in the PLOS ONE journal.

These are my main points of criticism, but on top of this, I find many other aspects of this paper problematic:

Abstract 

1- Abstract section is so general and does not have a scientific structure as a research paper.

The abstract has been revised, as detailed in recommendation No. 2.

2- The obtained result has not been appropriately shown, and the authors should mention the obtained results of applied methods, not general notes, in the abstract section.

The abstract has been revised. The revised abstract is shown below:

Abstract: The number of input factors affects the prediction accuracy of a model. Factor screening plays an important role as the starting point for data input. The aim of this study is to explore the influence of different factor screening methods on the prediction results. Taking the 2014 landslide inventory of Jingdong County as an example, a landslide database was constructed based on 136 landslide events and 11 selected factors, which were randomly divided into a training dataset and a test dataset according to a ratio of 7:3. Four factor screening methods, namely, the information gain ratio (IGR), GeoDetector, Pearson correlation coefficient and multicollinearity test (MT), were selected to screen the factors. A random forest (RF) model was then used in combination with each factor set for landslide susceptibility mapping (LSM). Finally, accuracy validation was performed using confusion matrices and ROC curves. The results show that factor screening is beneficial in improving the accuracy of the resulting model compared to the original model. Second, the IGR_RF model had the highest AUC value (0.9334), which was higher than that of the MT_RF model without factor screening (0.9194), and the IGR_RF model predicted the most landslides in the very high susceptibility zone (51.22%), indicating the good prediction performance of the IGR_RF model. Finally, the factor weighting analysis revealed that NDVI, elevation and aspect had the greatest influence on landslides in Jingdong County and that curvature had the least influence on landslides. This study can provide a reference for factor screening in LSM.

Introduction

In the first paragraph, the authors did not mention any references or statistics about landslides in the current study area and the rate of damages or losses of lives by this natural hazard.

A description of the landslide related to Jingdong County has been added to the first paragraph of the introduction. This is as follows:

Landslides are a common geological hazard in mountainous areas and are characterized by wide distributions, high frequencies of occurrence and rapid hazard formation [1]. Jingdong County is near the collision zone between the Indian Ocean plate and the Eurasian plate, exhibits active crustal movement, and is one of the key landslide prevention areas in Yunnan Province [2]. In 2008, a landslide and other natural hazards occurred in Taizhong township, Jingdong County, affecting 7,604 people in the township [3]. In 2016, 1 person died and more than 3,600 people were affected by landslides and other geological hazards in Jingdong County, with direct economic losses of approximately RMB 8.115 million yuan [4]. How to effectively prevent and reduce the occurrence of new landslide hazards is one of the main points of work for government departments today. Landslide susceptibility mapping (LSM) is a common method for landslide risk assessment [5]. LSM is the use of mathematical language to describe the probability of future landslides in an area under the influence of topography and geomorphology, geological structure, hydrology and meteorology, human activities and other predisposing factors, and it is used to map the distribution of landslide susceptibility [6,7]. LSM can provide a scientific basis for land planning, support major construction site selection and monitoring of landslide hazards, protect people’s lives and property effectively, and reduce unnecessary losses [8]. 

[1] Dou J, Yunus A P, Xu Y, et al. Torrential rainfall-triggered shallow landslide characteristics and susceptibility assessment using ensemble data-driven models in the Dongjiang Reservoir Watershed, China[J]. Natural Hazards, 2019, 97: 579-609. https://doi.org/10.1007/s11069-019-03659-4.

[2] Department of Natural Resources of Yunnan Province. Department of Natural Resources of Yunnan Province on the issuance of the 2020 Yunnan Province geological hazard prevention and control program. Department of Natural Resources of Yunnan Province. 2020 Nov 3 [Cited 2023 Aprial 20]. Available from: http://dnr.yn.gov.cn/html/2020/dizhizaihaifangzhi_1103/31197.html.

[3] Xinhua News Agency. Floods and landslides have affected 7,604 people in Taizhong Township, Jingdong County, Yunnan Province. Xinhua News Agency. 2008 Nov 4 [Cited 2023 June 23]. Available from: https://www.gov.cn/govweb/jrzg/2008-11/04/content_1139590.htm.

[4] Fu R. 1 person died and more than 3,600 people were affected by landslides in Jingdong County, Yunnan Province (many pictures). CNR News. 2016 Sep 21 [Cited 2023 Aprial 20]. Available from: http://news.cnr.cn/native/gd/20160921/t20160921_523150634.shtml.

[5] Merghadi A, Yunus A P, Dou J, et al. Machine learning methods for landslide susceptibility studies: A comparative overview of algorithm performance[J]. Earth-Science Reviews, 2020, 207: 103225. https://doi.org/10.1016/j.earscirev.2020.103225.

[6] Chung C J F, Fabbri A G. Validation of spatial prediction models for landslide hazard mapping[J]. Natural Hazards, 2003, 30: 451-472. https://doi.org/10.1023/B:NHAZ.0000007172.62651.2b.

[7] Guzzetti F, Galli M, Reichenbach P, et al. Landslide hazard assessment in the Collazzone area, Umbria, Central Italy[J]. Natural hazards and earth system sciences, 2006, 6(1): 115-131. https://doi.org/10.5194/nhess-6-115-2006.

[8] Cheng L, Li J, Duan P, et al. A small attentional YOLO model for landslide detection from satellite remote sensing images[J]. Landslides, 2021, 18(8): 2751-2765. https://doi.org/10.1007/s10346-021-01694-6.

The research background about the proposed subject and methods has not been appropriately mentioned, mainly applying approaches widely used in landslide susceptibility mapping in previous studies. Please update it with new references such as:

- https://doi.org/10.3390/f11080830

- https://doi.org/10.3390/f11040421

- https://doi.org/10.1016/j.catena.2018.01.012

I have revised the research background and research methodology and updated the references above.

Also, why did you not explain the novelty of the research adequately? Suppose authors utilized RF method implemented and assessed for LSM. Why did they not bring significant scientific resources in relation to the suggested model? Suppose the authors claim that the proposed methods are a new contribution to this research. In that case, they must bring defendable documents such as published scientific papers for their claim, clear it properly, and not bring the general notes found in any previously published papers.

The main goal of the paper should be better explained. It is not clearly stated. The goal should clearly and concisely explain the main scientific contributions of this work.

The "Introduction" section has been rewritten, please see the paper.

2.1. Study area

I did not see any description of the occurred landslides in the study area. Why do you choose this area for study? This section is a problem statement, and you should bring a logical reason for starting this research.

In accordance with the comments made in the introduction, a description of the landslides in the study area has been added to the first paragraph of the introduction and a partial description has been added to "2.1 Study area”. Please see the paper.

2.3. Factor data sources 

How do you select non-landslide locations? 

The selection of non-landslide sample points was based on the methodology of Wang et al [1]. It is described in section 2.2: In addition, 136 non-landslide units were randomly selected outside 500 m buffer zones around the landslide units.

[1] Wang Y, Wen H, Sun D, et al. Quantitative assessment of landslide risk based on susceptibility mapping using random forest and GeoDetector [J]. Remote Sensing, 2021, 13(13): 2625. https://doi.org/10.3390/rs13132625.

Figure 1: It is better separate landslide locations based on training and validation ratio.

I have modified the landslide samples in Figure 1 according to the training and testing scales.

Fig 1. The study area and landslides.

Figure 2: It is better to add landslide points on all factors. The fuzzy classification is not acceptable for conditioning factors. It is better use different classification methods such as natural breaks, Quantile and etc. Furthermore, you should add the names of all factors in figure captions.

We have added landslide points to all factors and updated the names of all factors in the figure captions.

For the classification method of factors, this paper firstly statistically analyses the spatial distribution of landslides in each factor, and then takes into account the spatial geographic information provided by landslides when using equal interval classification to reduce the uncertainty caused by zoning as much as possible. Taking the elevation as an example, no landslide occurs when the elevation is less than 950m, so we classify those less than 950 m into one category. In addition, according to Huang [1], it was found that the prediction accuracy was relatively good when the number of intervals was greater than 8. Combined with the range of values in the data, the number of classification intervals for most of our factors was also greater than 8 categories. The natural breakpoints (jenks) method was also considered at first, but the natural breakpoints (jenks) method is based on the values of the input data to classify, achieving the least variation within classes and the most variation between classes. The size and number of values will affect the size of the classification interval. Also, the method does not take into account the distribution of landslides, which to some extent increases or decreases the information provided by the different classifications.

[1] Huang F, Ye Z, Jiang S H, et al. Uncertainty study of landslide susceptibility prediction considering the different attribute interval numbers of environmental factors and different data-based models[J]. Catena, 2021, 202: 105250. https://doi.org/10.1016/j.catena.2021.105250.

Fig 2. Thematic map of landslide factors: (a) elevation, (b) slope, (c) curvature, (d) average annual precipitation (AAP), (e) normalized difference vegetation index (NDVI), (f) topographic wetness index (TWI), (g) distance to roads (DTRO), (h) distance to faults (DTF), (i) distance to rivers (DTRI), (j) aspect, and (k) lithology.

3. Methods

It is better to add applied equations for applied models by mathematical attitude for this section.

The "Methods" section has been revised and equations have been added. Please see the paper.

4. Results and Discussion

It is noted that a critical analysis of results has not been reported in the paper. The result section is so poor in writing and obtaining results. Also, the written results are not homogeneous due to no logical relationship between applied methods results.

Results and discussion have been separated and rewritten. In addition, relevant study details are compared with previous studies. See the 'Results' and 'Discussion' sections of the paper for details.

Figure 9: It is better to add landslide points on all parts of this figure.

Landslide points have been added in Figure 9.

Fig 9. The results of landslide susceptibility mapping (LSM) based on four different factor screening methods: (a) IGR_RF; (b) GD_RF; (c) PCC_RF; (d) MT_RF.

Reviewer #2: 

I think there are doubts issues about this manuscript following:

1. I don't think the objectives and the rationale of the study clearly stated.

Combined some previous studies and based on my experience, the performance of landslide susceptibility model (LSM) is related to the types of Mapping units (grid / slope units, Doi:10.1016/j.gr.2022.07.013.), and depends on algorithm selection (Doi: 10.1016/j.enggeo.2020.105972.), algorithm parameter optimization (Doi: 10.1016/j.geomorph.2020.107201.), Selection of samples (especially negative samples, Doi: 10.1007/s12583-020-1072-9), selection of conditioning factors (Doi: 10.1016/j.geomorph.2021.107623, Doi: 10.1016/j.gsf.2021.101211.), and scale / size of mapping units (Doi: 10.1007/s10064-021-02415-y, Doi: 10.1016/j.catena.2022.106428.). The importance of factor screening for LSM performance should be highlighted in the Introduction section. Meanwhile, It is strongly recommended to highlight the contribution of this paper in the "Introduction," "Discussion," or "Conclusion" sections.

The "Introduction", "Discussion" or "Conclusion" sections have been rewritten. The importance of factor screening for LSM performance has also been emphasized. See the individual sections of the paper for details.

2. I don't think there are sufficient analysis about previous literature in the section Introduction. In particular, there is a lack of some relevant literature in recent years on the screening conditioning factors of landslide susceptibility. I strongly recommends to refer the following previous literature in the section Introduction, and further clearly state the research status of the factor screening for LSM.

Liao et al., (2022), https://doi.org/10.1016/j.catena.2022.106428

Wen et al., (2023), https://doi.org/10.1016/j.jenvman.2023.118177

Wen et al., (2022), https://doi.org/10.1080/10106049.2022.2120547

Sun et al., (2022), https://doi.org/10.1016/j.gr.2022.07.013

Zhou et al., (2022), https://doi.org/10.1080/10106049.2022.2076928

Zhang et al., (2021), https://doi.org/10.1007/s10064-021-02415-y

Sun et al., (2021), https://doi.org/10.3389/feart.2021.713803

Zhou et al., (2021), https://doi.org/10.1016/j.gsf.2021.101211

Sun et al., (2021), https://doi.org/10.1016/j.geomorph.2021.107623

The status of research on factor screening in LSM has been further clarified in the 'Introduction' section, and references have been added.

3. Only 11 initial factors, which seems too few, and important conditioning factors could be missing for landslide susceptibility.

Jingdong County is located in the southwest of Yunnan Province, with complex topography and a backward economy, making it difficult to collect some of the data. Combined with the occurrence of geological hazards in Jingdong County, we carried out preliminary screening before carrying out landslide susceptibility mapping, and finally selected 11 factors.

4. It is not comprehensive to only use AUC value to access the LSMs’ performance and comprise the effects of different LSMs. It should be added the confusion matrix and the corresponding accuracy, precision, recall and F1-score. For example, even if the AUC value is very high, if the recall rate of landslide is low, it indicates that the model is not a good one.

The confusion matrix and the corresponding metrics have been added in the sections "3.3. Accuracy verification" and "4.2. Accuracy verification". The results are as follows:

The factors screened by the different factor screening methods were substituted into the RF model for landslide susceptibility modeling. The statistical results using the precision, accuracy, recall and F1 score are presented in Table 1. Among the precision indicators, IGR_RF and GD_RF showed the best results. In the accuracy metric, IGR_RF, GD_RF and PCC_RF showed the best results. In the recall and F1 metrics, PCC_RF results performed best. These results indicate that the predictive power of the models using factor screening (IGR_RF, GD_RF and PCC_RF) is improved compared to the results without elimination of factors (MT_RF), demonstrating the effectiveness of factor screening.

Table 1. Performance of different factor screening methods.

Method Precision Accuracy Recall F1 score

IGR_RF 0.8974 0.8780 0.8537 0.8750

GD_RF 0.8974 0.8780 0.8537 0.8750

PCC_RF 0.8780 0.8780 0.8780 0.8780

MT_RF 0.8537 0.8537 0.8537 0.8537

5. In fact, some researchers believe that factor screening is not important. It is strongly recommended to add RF-based LSM without screening factors and compare it with the LSMs’ performance under the 4 different factor screening methods.

As there was no multicollinearity between the factors in the multicollinearity test, all the factors were brought into the RF model. Therefore, this model was used as the comparison model without factor screening. The results of the multicollinearity test concluded that all factors met the requirements, which is similar to previous screening results [1-3], indicating that the MT model is insensitive to correlations between factors.

We have added relevant notes in 4.1 MT screening results, and 4.2 Accuracy validation and discussion.

[1] Chen W, Yang Z. Landslide susceptibility modeling using bivariate statistical-based logistic regression, naïve Bayes, and alternating decision tree models[J]. Bulletin of Engineering Geology and the Environment, 2023, 82(5): 190. https://doi.org/10.1007/s10064-023-03216-1.

[2] Liu Q, Tang A, Huang Z, et al. Discussion on the tree-based machine learning model in the study of landslide susceptibility[J]. Natural Hazards, 2022, 113(2): 887-911. https://doi.org/10.1007/s11069-022-05329-4.

[3] He Y, Zhao Z, Yang W, et al. A unified network of information considering superimposed landslide factors sequence and pixel spatial neighbourhood for landslide susceptibility mapping[J]. International Journal of Applied Earth Observation and Geoinformation, 2021, 104: 102508. https://doi.org/10.1016/j.jag.2021.102508.

Reviewer #3: 

The article employed IGR, GD, PCC, and MT methods for screening landslide conditioning factors and constrinfucted four landslide susceptibility models using the RF algorithm. The accuracy of these four methods was compared, and the research content of the article holds significance for landslide prevention and management. However, the article lacks substantial innovation, as the selected methods are commonly used in LSM. The discussion section is superficial, and certain sentences in the article are not coherent. It is recommended that the article undergo major revision and be reevaluated. Specific recommendations are provided below:

1. The author chose the RF algorithm because of its strong generalization ability and resistance to overfitting. However, based on my understanding, Boosting algorithms in ensemble learning, such as XGBoost and LightGBM, are better suited for addressing these issues.

Firstly, we thank the reviewers for their suggestions. For the Boosting algorithm in integrated learning, we conducted experiments using XGBoost and LightGBM and used a Bayesian optimization algorithm to perform hyperparameter search for both models. The ROC results are shown in the table 1. Overall, the RF algorithm gave the best results in this study area. No single model exists that is suitable for all regions, so the RF model was chosen.

Table 1. ROC curve results.

Factor screening Random forest XGBoost LightGBM

Information gain ratio 0.9334 0.9126 0.9027

GeoDetector 0.9256 0.8977 0.8914

Pearson correlation coefficient 0.9197 0.8917 0.8941

Multicollinearity test 0.9194 0.9001 0.9007

2. In my opinion, Random Forest consists of a large number of decision trees. Even if a factor has a minimal impact on landslide development in a specific area, the model will still learn this phenomenon. Please provide a detailed explanation of how redundant factors affect the accuracy of the model.

The section '5.1. Importance of factor screening' explains how redundant factors can affect the accuracy of the model, as follows:

Four factor screening methods (IGR, GD, PCC and MT) were selected for comparative analysis. The number of factors eliminated by IGR, GD and PCC was one, three and one respectively. In contrast, the results of the MT concluded that all factors met the requirements, indicating that the MT model is insensitive to correlations between factors, similar to the previous screening results [1,2]. We used the MT model as the original model without elimination of factors and analyzed it against the remaining three models, and found that the predictive power of the model implementing factor screening was significantly improved in LSM (Fig 8、Tables3 and 4). The AUC values for IGR_RF, GD_RF, and PCC_RF were 0.9334, 0.9256, and 0.9197, respectively, compared to 0.9194 for the original model without elimination. IGR_RF predicted five more landslides in the very high susceptibility zone compared to the original model. Secondly, within the very low susceptibility area, the original model predicted three landslides, while IGR_RF, GD_RF and PCC_RF were one, one and zero respectively. It can thus be seen that using more factors does not necessarily improve the prediction accuracy of the model, and redundant factors provide incorrect information to the prediction model. On the contrary, using the right factors can lead to better prediction results. Furthermore, given the same number of eliminated factors (IGR and PCC models), IGR obtained better prediction results because PCC analyses the correlation and redundancy between factors and does not consider the relationship between factor attribute characteristics and landslides. The IGR, on the other hand, is based on the probability of landslides occurring under different factor conditions and takes into account the relationship between factor attribute characteristics and landslides. Overall, the IGR_RF model predicts better results.

[1] He Y, Zhao ZA, Yang W, Yan H, Wang W, Yao S, et al. A unified network of information considering superimposed landslide factors sequence and pixel spatial neighbourhood for landslide susceptibility mapping. International Journal of Applied Earth Observation and Geoinformation. 2021; https://doi.org/10.1016/j.jag.2021.102508

[2] Liu Q, Tang A, Huang Z, Sun L, Han X. Discussion on the tree-based machine learning model in the study of landslide susceptibility. Natural Hazards. 2022; 113(2):887-911. doi: 10.1007/s11069-022-05329-4.

3. It is suggested to include information about rainfall and vegetation coverage in the overview of the study area.

Information on rainfall and vegetation cover has been added to the study area. See section 2.1: The average annual rainfall and annual temperature are 1086.7 mm and 18.3 ℃ respectively. It has distinct dry and wet seasons. Jingdong County is the only county in China to have two national nature reserves - the Wuliang Mountain and Ailao Mountain National Nature Reserves - and is rich in forest resources, with a county-wide forest coverage rate of 77.01%.

4. In Table 1, the article states that the grid resolution is 30m. The resolution of NDVI and elevation is also 30m, which is acceptable. However, the resolution of the rainfall data mentioned in the article is 1km. Inconsistent resolutions can significantly impact the accuracy of the model.

Jingdong County is located in southwestern Yunnan Province, with complex topographic conditions, low population density and few meteorological stations. Combining characteristics such as data accessibility and quality of acquired data, 1km resolution rainfall data were selected. To facilitate later calculations and analysis, we resampled the 1km resolution data to 30m, which is consistent with the data taken in some studies [1-4].

[1] Wang Z, Xu S, Liu J, et al. A Combination of Deep Autoencoder and Multi-Scale Residual Network for Landslide Susceptibility Evaluation[J]. Remote Sensing, 2023, 15(3): 653. https://doi.org/10.3390/rs15030653.

[2] Liao M, Wen H, Yang L. Identifying the essential conditioning factors of landslide susceptibility models under different grid resolutions using hybrid machine learning: A case of Wushan and Wuxi counties, China[J]. Catena, 2022, 217: 106428. https://doi.org/10.1016/j.catena.2022.106428.

[3] Yuan X, Liu C, Nie R, et al. A Comparative Analysis of Certainty Factor-Based Machine Learning Methods for Collapse and Landslide Susceptibility Mapping in Wenchuan County, China[J]. Remote Sensing, 2022, 14(14): 3259. https://doi.org/10.3390/rs14143259.

[4] Wang B, Lin Q, Jiang T, et al. Evaluation of linear, nonlinear and ensemble machine learning models for landslide susceptibility assessment in Southwest China[J]. Geocarto International, 2022: 2152493. https://doi.org/10.1080/10106049.2022.2152493.

5. Why was reclassification performed? Additionally, most studies use the natural breakpoints (jenks) method for reclassification. Why did the author choose equal intervals for reclassification?

Some methods (GeoDetector and Information Gain Ratio) require input of discrete variables, so continuous variables (elevation, slope, rainfall et al.) have to be reclassified.

There are many methods of factor grading, such as the common natural breakpoints (jenks) method and equal interval method. Different classification methods can lose the true information of landslides to different degrees, and it has been a technical challenge for many scholars to study how to maximize the retention of true information of landslides. The natural breakpoints (jenks) method is based on the classification of the values of the input data to achieve the minimum variation within a class and the maximum variation between classes. The size and number of values will affect the size of the classification interval. Although most studies use the natural breakpoints (jenks) method, the method does not take into account the distribution of landslides, which to some extent increases or decreases the information provided by the different classifications. In this paper, the spatial distribution of landslides in each factor is first statistically analyzed, and then the spatial geographic information provided by landslides is taken into account when using equal interval classification to reduce the uncertainty caused by zoning as much as possible. Taking elevation as an example, no landslides occur when the elevation is less than 950m, so we classify those less than 950m into one category. However, the results using the natural breakpoints (jenks) method will not take this into account (Fig 1). In addition, according to Huang [1], it was found that prediction accuracy is relatively good for interval numbers greater than 8. Combined with the range of values in the data, the number of classification intervals for most of our factors is also greater than 8 classes.

Fig 1. Schematic diagram of the classification results using the natural breakpoints (jenks) method (with elevation as an example). 

[1] Huang F, Ye Z, Jiang S H, et al. Uncertainty study of landslide susceptibility prediction considering the different attribute interval numbers of environmental factors and different data-based models[J]. Catena, 2021, 202: 105250. https://doi.org/10.1016/j.catena.2021.105250.

6. None of the four methods used by the author for screening were subjected to a significance test. Results without a significance test lack persuasiveness. Please include the results of the significance test.

Thank you for your comments on my manuscript. We have previously considered significance tests. However, considering the principle of significance testing and the objective reality of landslides, we did not conduct significance testing for the following reasons:

Significance tests are more suitable for tests where there is a linear relationship between factors. In contrast, landslide formation conditions are complex, and there are mostly non-linear relationships with the various factors. This is the reason for the increasing popularity of machine learning models as opposed to traditional mathematical and statistical models. In addition, significance tests assume that the data are normally distributed. However, for landslides, the landslide factor data involved are usually characterized by non-normal distribution due to factors such as geography and human activities. Therefore, instead of conducting significance tests, we verified the accuracy of the results by means of confusion matrices and ROC curves.

7. In lines 292-296, what criteria were used to determine strong correlation? The correlation coefficient between elevation and rainfall is 0.67, indicating a strong correlation. Why wasn't it eliminated?

The criteria here refer to previous studies [1-3], where we consider two factors to be strongly correlated when the absolute value of the correlation coefficient between the factors exceeds 0.7. The correlation coefficient between altitude and rainfall did not exceed 0.7 and all were not eliminated. References have been added to the methods and results.

[1] Dou J, Yunus A P, Merghadi A, et al. Different sampling strategies for predicting landslide susceptibilities are deemed less consequential with deep learning[J]. Science of the total environment, 2020, 720: 137320. https://doi.org/10.1016/j.scitotenv.2020.137320.

[2] Wu Y, Ke Y, Chen Z, et al. Application of alternating decision tree with AdaBoost and bagging ensembles for landslide susceptibility mapping[J]. Catena, 2020, 187: 104396. https://doi.org/10.1016/j.catena.2019.104396.

[3] Martín B, Alonso J C, Martín C A, et al. Influence of spatial heterogeneity and temporal variability in habitat selection: A case study on a great bustard metapopulation[J]. Ecological Modelling, 2012, 228: 39-48. https://doi.org/10.1016/j.ecolmodel.2011.12.024.

8. It is recommended to divide the results and discussions into two sections. The discussion section should delve deeper into the relationship between factors and landslide mechanisms.

As required, we have divided the results and discussion into two parts. The mechanisms by which the main factors affect landslides are discussed in the section '5.2 Influence of the screened factors on landslides'.

9. Figure 8: The author only compared the accuracy of the four factor methods combined with the RF model, which is incomplete. To enhance the persuasiveness of the article, the author should include a comparative model - an RF model without factor screening.

As there was no multicollinearity between the factors in the multicollinearity test, all the factors were brought into the RF model. Therefore, this model was used as the comparison model without factor screening. The results of the multicollinearity test concluded that all factors met the requirements, which is similar to previous screening results [1-3], indicating that the MT model is insensitive to correlations between factors.

We have added relevant notes in 4.1 MT screening results, and 4.2 Accuracy validation and discussion.

[1] Chen W, Yang Z. Landslide susceptibility modeling using bivariate statistical-based logistic regression, naïve Bayes, and alternating decision tree models[J]. Bulletin of Engineering Geology and the Environment, 2023, 82(5): 190. https://doi.org/10.1007/s10064-023-03216-1.

[2] Liu Q, Tang A, Huang Z, et al. Discussion on the tree-based machine learning model in the study of landslide susceptibility[J]. Natural Hazards, 2022, 113(2): 887-911. https://doi.org/10.1007/s11069-022-05329-4.

[3] He Y, Zhao Z, Yang W, et al. A unified network of information considering superimposed landslide factors sequence and pixel spatial neighbourhood for landslide susceptibility mapping[J]. International Journal of Applied Earth Observation and Geoinformation, 2021, 104: 102508. https://doi.org/10.1016/j.jag.2021.102508.

10. It is advisable to present the conclusions in bullet points.

We have presented the conclusions in bullet points. These are as follows:

For machine learning models, the quality of the input factor data affects the prediction accuracy of the model. Factor screening plays an important role as the starting point for data input. To compare and analyze the effects of different factor screening methods on the prediction results, four factor screening methods – IGR, GD, PCC and MT – were used in this study. The 2014 landslide hazard data from Jingdong County, Yunnan Province, were used as an example, and the slope unit was used as the basic evaluation unit. Each of the four factor screening methods was combined with the RF model for susceptibility assessment. The results of the study showed the following:

(1) Compared with the prediction results without eliminating factors, factor screening is beneficial to improving the prediction accuracy of the model, and there are some differences in the prediction results among the different screening methods. IGR and GD consider the relationships between factors and landslides and can calculate factor weights. PCC and MT do not calculate the weights of factors.

(2) The results of the ROC curve analysis show that the IGR_RF model has a higher AUC value (0.9334) than the other models (GD_RF, PCC_RF, MT_RF). The IGR_RF model predicted the most landslides in the very high susceptibility area.

(3) Based on the results of the factor screening with the IGR and GD methods, the top 3 factors are NDVI, elevation and aspect. Curvature has the least influence on landslides. The influence of the main factors on landslides was analyzed in relation to the actual situation in Jingdong County. Landslides mainly occur in areas with NDVI values of 0.4-0.6, elevations of 950 m-2200 m and a west-facing orientation.

In summary, IGR_RF predicts the best results. IGR is calculated based on the probability of landslides occurring under different factor conditions, which considers the relationship between factor attribute characteristics and landslides and improves the reliability of the input data. In addition, the method can give the weight values of different factors to provide a reference for landslide prevention and management. It provides a reference for other scholars in factor screening.

11. What specific aspect does the factor "Rainfall" refer to in the article? Is it the average annual rainfall or something else? Please clarify this point.

The 'Rainfall' in the paper refers to the average annual precipitation, and we have amended the statement in the article and in the figures.

---

## [Decision Letter · Decision Letter 1]

2 Oct 2023

Study on landslide susceptibility mapping with different factor screening methods and random forest models

PONE-D-23-12592R1

Dear Dr. Li

We’re pleased to inform you that your manuscript has been judged scientifically suitable for publication and will be formally accepted for publication once it meets all outstanding technical requirements.

Kind regards,

Salim Heddam

Academic Editor

PLOS ONE

Additional Editor Comments (optional):

Reviewer 1:I believe, that in its present form, your article is appropriate to publish in the PLOS ONE journal and will be interesting for the readers.

Reviewer 2:Dear authors, thank you for your response to the previous round of review comments. I think all my comments have been addressed.

Reviewers' comments:

Reviewer's Responses to Questions

**Comments to the Author**

1. If the authors have adequately addressed your comments raised in a previous round of review and you feel that this manuscript is now acceptable for publication, you may indicate that here to bypass the “Comments to the Author” section, enter your conflict of interest statement in the “Confidential to Editor” section, and submit your "Accept" recommendation.

Reviewer #1: All comments have been addressed

Reviewer #2: All comments have been addressed

2. Is the manuscript technically sound, and do the data support the conclusions?

Reviewer #1: Yes

Reviewer #2: Yes

3. Has the statistical analysis been performed appropriately and rigorously? 

Reviewer #1: I Don't Know

Reviewer #2: Yes

4. Have the authors made all data underlying the findings in their manuscript fully available?

Reviewer #1: Yes

Reviewer #2: Yes

5. Is the manuscript presented in an intelligible fashion and written in standard English?

Reviewer #1: Yes

Reviewer #2: Yes

6. Review Comments to the Author

Reviewer #1: Dear Authors,

I believe, that in its present form, your article is appropriate to publish in the PLOS ONE journal and will be interesting for the readers.

Reviewer #2: Dear authors, thank you for your response to the previous round of review comments.

I think all my comments have been addressed.

7. PLOS authors have the option to publish the peer review history of their article (what does this mean?). If published, this will include your full peer review and any attached files.

Reviewer #1: No

Reviewer #2: No

---

## [Editor Report · Acceptance letter]

4 Oct 2023

PONE-D-23-12592R1 

Study on landslide susceptibility mapping with different factor screening methods and random forest models 

Dear Dr. Li:

I'm pleased to inform you that your manuscript has been deemed suitable for publication in PLOS ONE. Congratulations! Your manuscript is now with our production department. 

Kind regards, 

on behalf of

Dr. Salim Heddam 

Academic Editor

PLOS ONE